# Protein Function Analysis through Machine Learning

**DOI:** 10.3390/biom12091246

**Published:** 2022-09-06

**Authors:** Chris Avery, John Patterson, Tyler Grear, Theodore Frater, Donald J. Jacobs

**Affiliations:** 1Department of Bioinformatics and Genomics, University of North Carolina at Charlotte, Charlotte, NC 28223, USA; 2Department of Physics and Optical Science, University of North Carolina at Charlotte, Charlotte, NC 28223, USA

**Keywords:** machine learning, protein structure prediction, protein–protein interactions, protein dynamics, protein function, allostery, conformational sampling, force fields, molecular docking

## Abstract

Machine learning (ML) has been an important arsenal in computational biology used to elucidate protein function for decades. With the recent burgeoning of novel ML methods and applications, new ML approaches have been incorporated into many areas of computational biology dealing with protein function. We examine how ML has been integrated into a wide range of computational models to improve prediction accuracy and gain a better understanding of protein function. The applications discussed are protein structure prediction, protein engineering using sequence modifications to achieve stability and druggability characteristics, molecular docking in terms of protein–ligand binding, including allosteric effects, protein–protein interactions and protein-centric drug discovery. To quantify the mechanisms underlying protein function, a holistic approach that takes structure, flexibility, stability, and dynamics into account is required, as these aspects become inseparable through their interdependence. Another key component of protein function is conformational dynamics, which often manifest as protein kinetics. Computational methods that use ML to generate representative conformational ensembles and quantify differences in conformational ensembles important for function are included in this review. Future opportunities are highlighted for each of these topics.

## 1. Introduction

Our review presents important questions about proteins and their functions that have been around for several decades in terms of molecular biology, biochemistry, and biophysics. This scientific knowledge is where practiced researchers have an advantage; however, there is a major difference in how machine learning (ML) is used in computational biology today compared to how it was used 20 years ago [1]. There is now a convergence between the two fields as illustrated in Figure 1. This new way of thinking puts computational biology at a stage where problems that were unsolvable by old methods can become solvable, or at least limited solutions can provide increased accuracy and/or speed by incorporating ML methods. The promise that ML can help solve challenging problems applies to all areas of science, engineering, and beyond (e.g., medicine, art, music, economics, public policy) [2,3,4].

In this review, we make no distinction between old or new ML methods, or whether the ML method is based on clustering, decision trees, support vectors, projection pursuit or deep/shallow neural networks. If any form of ML has been used to help elucidate protein function, we mention the method if we found it in our literature searches. For those ML methods that escaped our attention, this was not intentional, and we regret our oversight. We also discuss experimental data when they are a critical component of a method in computational biology, such as the prediction of protein structure. We discuss limited aspects of experimental methodology only if this process is part of a method within computational biology. Not all aspects of computational biology are covered in equal depth, either because of our bias toward what we believe is the most interesting aspects of protein function, or because of what is available in the computational biology literature. The biases we have towards the topics covered in this review are unabashedly applied to help researchers fuse structural bioinformatics and computational biology with ML.

The rapid introduction of novel ML methods has already transformed the field of computational biology [5]. Experts in the field and early doctoral students are on close footing regarding development of new methods that use ML at a fast pace, which is not expected to slow down soon. This environment may foster many successful handshakes between old and new schools of thought, creating a breeding ground for new ideas. We believe it would be a mistake to promote a particular methodology that uses ML without performing an independent and extensive fair comparison test between methods. This is often carried out in the field of computational biology with certain blind competitions, which we cite when available. Besides making the scope of the review too large by including technical and controlled head-to-head comparisons, the reader should be aware that changes are being made so quickly that a research group will take ideas from others and create a competitive scheme in turn-key fashion. With easy-to-use ML tools and rapid developments in artificial intelligence (AI), this surge in high-intensity research is beneficial. Nevertheless, there are two frustrating elements that will occur before things start to settle down. Firstly, not all good ideas will survive, and not all ideas that survive will be explainable. Secondly, it is unfortunate that not all results will be reproducible, because researchers generally do not always precisely define the methodology and specify critical details. One way to mitigate these problems is to set standard datasets for training models to establish controls and benchmarks. This is happening in some areas of computational biology and ML [6] that we briefly mention.

Given the emergence of a new playing field that crosses ML with computational biology, we have set out to answer this two-part question: what is the current state of the art, and how does it further our knowledge on protein function? By reading the literature, we have answered this question fairly robustly. We hope that this review will help others as much as it has helped us understand the changing landscape in computational biology in terms of protein function analysis using ML. Section 2 of the paper highlights the biophysical aspects of protein function and presents a brief historical overview of how computational biology and machine learning have converged. Section 3 then discusses a wide range of computational biology applications using ML. We conclude in Section 4, where we highlight the remaining challenges and suggest future opportunities.

## 2. Foundational Concepts of Protein Function

The biochemical activity of life is orchestrated by proteins, polymer chains of amino acid molecules called residues. Their function can be autonomous or in concert, as multiple-chain complexes. They play a critical biological role by conducting interactions and controlling a myriad of processes in the crowded molecular environment of cells. The in vivo environment consists of a large and diverse population of proteins mixed with other molecules that include osmolytes, ions, fatty hydrocarbons, and other macro- and meso-molecules. This large mixture of molecules within cells and at other locations of a living organism, such as the extracellular matrix, continuously undergoes chemical reactions driven by thermodynamic and kinetic considerations. To maintain reliability in the dynamically changing network of chemical reactions, proteins evolved to maintain a functionally relevant conformational ensemble that satisfies thermodynamic and kinetic stability conditions [7]. Protein function builds on the physical and chemical aspects of its environment that can drive specificity for a particular process. This specificity is encoded in the primary structure to form the basis for the central dogma of biology and the sequence–structure–function paradigm in structural biology. However, the sequence–structure–function paradigm is shallow, analogous to the title of a story, whereas the description of protein function requires a novel for each protein.

Christian Anfinsen and co-workers showed that the 3D structure of proteins is due to thermodynamics [8]. Thermodynamics spontaneously drive the protein into a stable, free-energy basin that is consistent with the environment. In proteins, structure rather than sequence tends to be conserved among proteins that perform the same function, even in proteins that are analogous across species [9,10]. From this perspective, a protein’s 3D structure is arguably the most important physical attribute of a protein. Nevertheless, a single structure is a representation of an ensemble of conformations. Indeed, an intrinsically disordered protein (IDP) has an ensemble of conformations that cannot be accurately represented by a single native 3D structure [11]. In general, a protein is rigid when its conformational ensemble acutely preserves a native 3D structure, or perhaps it is so flexible that there is no conserved single domain that can be identified within the conformational ensemble. There is a broad spectrum of conformational characteristics that bridge these two extremes.

The common characteristic responsible for protein function is how it interacts with partners [12,13,14]. The strength of a protein’s interaction with other proteins or types of molecules can range from weak to strong, depending on its sequence and environment. The interaction between different partners is modulated by the environment, which can include the effects of post-translational modulation on a protein, such as glycosylation, gradients of solutes or solvents, or state changes such as pressure, pH, or temperature. The environment controls a large network of intra- and inter-molecular interactions that strongly depend on molecular concentrations. Although observed properties of a protein depend on its environment, it is common to focus on intrinsic properties where environmental effects are largely ignored. This approach is fruitful because the conditions for most life on planet Earth are constrained. As water is essential for life as we know it, organisms have evolved to live in extreme conditions such as in hot springs or deep within the ocean, but generically operate under similar molecular constraints. This can be observed for a given family of proteins from organisms living in diverse environments that share the same biomolecular function [15].

In general, proteins that are members of the same functional family lose their function when they are placed in conditions too different from their native environment. When comparing proteins in their respective native environments, we find that they tend to have similar intrinsic properties [16,17,18]. A careful analysis shows that proteins within the same functional family have large segments of their sequence conserved. This can be determined by aligning protein sequences using bioinformatic sequence alignment methods [19,20,21]. Multiple sequence alignment reveals corresponding amino acids between proteins when accounting for the possibility of mutations, insertions, and deletions [22,23].

As illustrated in Figure 2 the important components for quantifying protein function are sequence, structure, stability, dynamics and knowledge of the partners with which a protein interacts: ligands, ions, membranes and other proteins. A ligand can be a protein, and protein–protein interactions are an important aspect of protein function that is central in systems biology [24,25]. When applied to understanding the characteristics of proteins and their functions, computational biology rarely attempts to model all the effects described above at the same time. Different aspects of protein function are usually considered separately, and approximations are inevitably made [26,27]. This review examines how ML is used to better quantify protein function, and takes into account aspects of sequence, structure, dynamics and binding.

### 2.1. Elucidating Protein Function Using Computational Biology: A Short History

In the early days of computational biology, upon the advent of the digital computer, the main interest was using computers to help with taxonomy, where similar traits classify organisms. Many experimental comparative studies at the molecular level of proteins were also conducted, which helped establish the idea of molecular evolution [28]. The emphasis gradually shifted to genetics and evolutionary relationships found in DNA sequences, especially after the introduction of dynamic programming for alignments presented by Needleman and Wunsch [29].

In the 1970s, structural considerations of proteins began to be of interest. As it became easier to solve the inverse problem for structure determination using X-ray crystallography, protein structures became more readily available. In 1971, the protein data bank (PDB) was initiated with seven protein structures [30]. The forethought for building such a database is remarkable, as it has been an indispensable resource for serving as the backbone of structural bioinformatics and computational biology.

Computationally addressing how proteins dynamically fold also started in the 1970s [31], although computer power and molecular models were limited. In the late 1980s and throughout the 1990s, simplified bead models, including simulations on lattices [32], were used to understand the basic physics of the protein folding process. It was also common to consider unfolding from native states, expanding upon the Go model [33], which also increased the need to know about more protein structures. In the early 1990s, it became clear that accurate prediction of protein structure is essential when it comes to understanding protein function, even if the dynamic protein folding pathways remain unknown. Protein function began to be simulated in the late 1990s once molecular dynamics (MD) of proteins exhibited numerical stability over long timescales (approaching milliseconds with large computing resources). These computational studies required starting with a known protein structure from the PDB.

As attention began to shift to native state dynamics of globular proteins, the mechanisms of action underlying protein function were able to be studied computationally. This included mutation studies and monitoring changes in dynamics of proteins as certain residues mutated (usually on timescales of 1 μs or less). Native state dynamics also stimulated interest in boosting the accuracy of dynamic allostery models [34]. This capability naturally lead to the development of models and algorithms for computational protein design and protein stability prediction. Unfortunately, MD simulation is not an appropriate tool for calculating protein stability because of its inability to properly generate ensembles large enough to calculate thermodynamic properties, not to mention approximations in force fields.

To avoid brute force MD simulation, thermodynamic models that assumed additivity in free-energy components were proposed to rapidly calculate thermodynamic quantities. Unfortunately, these simple additive models failed miserably [35] due to the non-additivity of conformational entropy [36]. However, in the early 2000s, with proper accounting for molecular constraints using rigidity theory, the non-additivity found in conformational entropy was accurately incorporated using a distance constraint model (DCM) [37]. For example, the DCM accurately described the heat capacity of proteins [38,39,40], including cold denaturation, because the model accounts for solvent effects [41]. From the DCM, quantitative stability and flexibility relationships are determined for proteins [16], which has been useful for understanding protein evolution and helps with protein design [42,43,44].

In tandem with interest in native state dynamics, starting in the mid 1990s, there was a strong interest in docking small molecules to proteins, characterize binding sites in proteins, and estimate binding affinities through computational means. Many docking methods were created, but once again these methods lacked the proper conformational sampling that considers the flexibility of the protein and the ligand. In addition, additive models for free-energy shifts used in scoring functions generally ignored entropic contributions to binding. For these reasons, developing methods for accurate molecular docking has been a major challenge in computational biology. The need for efficient and accurate docking methods in drug discovery drove the development of such methods, which led to large databases of potential small-molecule drugs, and myriad algorithms that combined data mining with quantitative structure–activity relationships (QSAR) [45].

These days, a systems biology approach is popular for tracking protein–protein interactions, which is critically important for biological regulation and maintaining homeostasis. However, predicting protein–protein interactions involves a docking problem that is particularly difficult because of the innate flexibility and size of binding regions on proteins, as well as the accuracy of scoring functions. This topic represents a current challenge in computational biology.

### 2.2. Convergence of Machine Learning and Computational Biology

Along with computational biology, ML has its own interesting historical development. Initially, methods for discriminant analysis were developed in statistics that are now considered ML [46,47]. However, the term “machine learning” was first coined by Arthur Samuel, who developed a computer program to play checkers in the early 1950s [48,49]. Frank Rosenblatt coined the term “perceptron” by combining the ideas of neural biology and optimization of game-playing objectives introduced by Samuel [50]. Over the years, artificial neural networks have gained popularity due to better algorithms and faster/larger computers. Especially with GPUs, deep learning with several or more perceptron layers is tractable. Moreover, the concept of rectifying unit or activation function has generalized and replaced the original perceptron concept.

At present, many ML methods are linked to statistical analysis, with optimization through automated procedures to draw statistical conclusions. The domain of ML is often divided into three main categories: unsupervised, supervised and reinforcement learning. Unsupervised and supervised learning uses data to train a model to perform a particular task. Unsupervised learning methods aim to learn patterns within a dataset without additional categorical information on how the data are structured. Supervised learning uses additional information during the training, such as class labels. Reinforcement learning takes the approach of self-learning, in which an agent learns to perform a particular action through trial and error, guided by a reward system, to perform a particular task when triggered by certain environmental conditions.

Other frameworks of ML have emerged, such as semi-supervised learning, which uses both labeled and unlabeled data to learn a particular task. The goal of ML is to train a model to perform a useful action or task. Some common functions of ML algorithms include: clustering, binary and multi-class classification, regression, generative modeling, natural language processing (NLP) and dimensionality reduction (DR). Table 1 provides an overview of general ML algorithms, along with where they fall in the landscape of ML.

Table 1 demonstrates the breadth of different types of algorithms which make up the current landscape of the ML field. This landscape is constantly being sculpted by new developments and, as such, is constantly changing. Currently, method development has accelerated dramatically in many areas of ML, such as generative modeling and NLP, which has made its way into the latest methods in computational biology. Generative modeling refers to algorithms which are able to learn patterns from the training data and use these patterns to create new data points from scratch. These methods have been used extensively in application domains such as image analysis. The variational autoencoder (VAE) [51] is an example of a generative model. VAEs consist of two networks that learn to encode samples to a low-dimensional space and decode (reconstruct) samples from the low-dimensional space, respectively. After training, the latter network can be used independently to construct new samples. VAEs are widely used in computational biology for dimension reduction, as well as its generative function. Generative Adversarial Networks (GAN) [52] are another example of popular generative models where the model consists of a generator and a discriminator. The generator constructs random samples, while the discriminator tries to identify which of two samples presented to it are real. The two are trained in opposition to each other, where the generator learns to trick the discriminator by generating more realistic samples. The VAE and GAN generative models have recently made their way into computational biology applications.

Natural language processing (NLP) is popular in ML research, where the goal is to analyze and generate sequence data such as text analysis/speech recognition or AI-powered chatbot models. Traditionally, recurrent models such as LSTM [53] have been used for these tasks because they inherently treat the data as a sequence. More recently, transformer networks are used in highly popular models such as GPT-3 or BERT [54,55] for text generation. Transformer networks have made several advancements in sequence models [56]. In particular, the inclusion of attention mechanisms allowed the model to put emphasis on the order of inputs, providing context and long-range correlations that can be exploited in an input sequence. NLP models are a popular choice in computational biology with respect to data-mining protein and DNA sequences.

Another model gaining significant traction in both the ML and computational biology communities is graph neural networks (GNN) [57]. These models treat the data as a graph, or a set of nodes that each contain a vector of features connected by edges according to a topology. For example, the nodes may represent atoms in a molecule and the edges defined according to the bond structure. A mechanism called message passing allows information to flow across the edges of the graph according to a set of message-passing rules, updating the feature representation at the nodes each time messages are passed. This allows the model to learn deeply encoded representations of data structure, which can then be used in a top-level model that performs a task. End-to-end training allows the GNN to learn representations specific to the task being performed. Graph convolutional neural networks (GCNN) have been shown to be very effective for molecular fingerprinting [58,59] and are being explored in a wide array of applications in computational biology.

Although ML addresses some of the limitations of traditional methods in computational biology, it also has its own caveats. Many methods require large amounts of data to train the model. When the data are high dimensional, a preprocessing step of DR is often required. As the amount of data and variables analyzed increases, algorithm performance typically wanes due to the curse of dimensionality [60]. Another problem is over-fitting to random fluctuations present in training data, or under-fitting due to incomplete sampling, which is associated with lower variance in the training sub-sample compared to the population sample. Protein function analysis often deals with high-dimensional data and low statistics, which push ML to its limits. We are now seeing a convergence between ML and computational biology, because the field of ML is now addressing the sampling problems encountered in computational biology. Of course, advances in ML are driven by applications. This convergence will help advance computational biology as much as ML. We are at the beginning of a new chapter in computational biology, meaning many gold-standard algorithms will become obsolete in the near future.

## 3. Selected Applications of Machine Learning in Computational Biology

In this section, we will focus on four pillars of computational biology, as shown in Figure 2. Although well known, but often neglected in the way one visualizes protein function, the thematic connection we make through this section is conformational ensembles, and how ML can help advance computational biology by providing better low-dimensional representations of the characteristics of these ensembles.

### 3.1. Protein Structure Prediction

The vast number of functions performed by proteins are facilitated by their 3D structure. Accurate protein structure prediction (PSP) opens the door to engineering novel protein functions in medicine, furthering our understanding of mechanisms through simulation and experimentation. Historically, the two most important computational approaches to PSP are template-based and template-free modeling. Template-based modeling is based on the principle that similar amino acid sequences share the same folds [61,62,63]. As a result, known empirical structures can be used to model unknown structures: homology modeling. The protein structure initiative (PSI) of the early 2000s aimed to distribute 3D structural information for naturally occurring proteins through experimental methods such as: (i) X-ray crystallography (XRC); (ii) nuclear magnetic resonance (NMR); (iii) cryo-electron microscopy (cryo-EM), each with its advantages and disadvantages [64,65]. Other available experimental methods include SAXS, CD, FRET, and Raman spectroscopy [66]. Notably, many protein structures are resolved when bound to a ligand. Some protein structures are obtained without a bound ligand, and some proteins have many structures, each with a different ligand. This empirical information is extremely useful for training and testing the docking software [67] often needed in drug design.

The research community religiously deposits experimentally resolved protein structures (and other non-protein macro molecular structures) into the PDB, where these data are used to improve the accuracy of template-based modeling. Template-free modeling generates protein structures without the use of homologs in the PDB [68]. This method typically relies on physics-based energy functions and is much more computationally intensive than template-free modeling. Knowledge-based approaches dominate the field, yielding deterministic designs for stable proteins or homology-based structural predicts [69,70,71]. Of course, there are algorithms that combine the these approaches. With ML, the lines between template-based and template-free modeling have become more seamless, and combinations of the two approaches are more commonplace [62,72].

The critical assessment of methods of protein structure prediction (CASP) is a biennial competition where the amino acid sequences of experimentally determined structures are provided to participants through double-blind assignment. The experimental structures used in CASP competitions are initially not deposited into the PDB until after the competition. Participants are expected to submit predicted models using their unique method of determination. Models are evaluated using the Z-scores of backbone conformational similarity using various measures for accuracy. The most successful methods over the last few rounds of CASP are given in Table 2 and represented in Figure 3 [73]. Although ML has made great strides in the field of PSP there are methods such as Feig-R2, which used physics-based refinement through MD simulations [74]. The Feig-R2 method was competitive against the ML methods from the Baker and Zhang groups.

It is important to use ML whenever it enhances computational efficiency, but ML need not dominate an approach, nor turn everything into a black box model. However, having said this, we see ML methods such as AlphaFold and AlphaFold2 have produced substantially better results. Researchers should be aware that this new paradigm does not make other methods for PSP obsolete. CASP is a competition, but more importantly, CASP provides a conduit for PSP experts worldwide to share ideas. In our view, ML is a powerful multifaceted tool, but ML will not always be the best tool that solves a problem. Ideally, ML should be used to enhance models rooted in basic physics and chemistry principles.

Debuting at CASP-Round XIV, AlphaFold2 by Google Deep Mind has raised the bar in PSP by achieving prediction accuracy near the experimental limits, and scoring in the competition far higher than their competitors [84]. AlphaFold2 is a completely reworked model of the AlphaFold method presented at CASP13 [73,89]. The success of AlphaFold2 comes from the unique implementation of deep neural networks and evolutionary history via computed MSA [72,84,90,91,92]. The high accuracy of AlphaFold2 should not be seen as a sign that we are close to the endgame for PSP. Rather, the success of AlphaFold2 points to new horizons to explore, shown by similar works such as RoseTTAfold [93] or an alphafold reproduction such as openfold [94]. Naturally, many areas of improvement are needed, such as accounting for the role of the environment or predicting structures without the need for multiple sequence alignments (MSAs), as seen in OmegaFold [95]. A new chapter has begun, and programs such as openFold [94] and RoseTTAfold [93] are emerging. OpenFold is a recreation of AlphaFold using pytorch and RoseTTAfold is a “three track” neural network [93,94].

It cannot be overemphasized that once structures are determined, mechanisms of action can be better understood by other computational methods. With a model structure in hand, new therapeutics can be developed much more rapidly, often enhanced by computer-aided structure-based drug discovery (SBDD) methods [96,97]. The effectiveness of SBDD has been enhanced by ML as well as discussed in [98,99]. Of course, many proteins cannot be crystallized because they have intrinsically disordered regions [100]. Sometimes only parts of proteins can be crystallized, as there are proteins with stable native structure combined with disordered regions. This raises a more general question of what a model structure means. The crystal structure represents a snapshot of the protein in an environment different from in vitro and in vivo conditions. In solution, proteins explore many conformations, and those that are accessible are called an ensemble. Structure prediction must shift to predicting distributions that model conformational ensembles, which is discussed next.

### 3.2. Conformational Ensembles

The native state of a protein usually refers to a folded structure, which is a functionally relevant conformation of the molecule within the global minimum of the free-energy surface (FES). From the perspective of the free-energy landscape, the native state of a protein corresponds to a native free-energy basin associated with a native conformational ensemble. Zooming into the native free-energy basin at high resolution generally reveals multiple functionally relevant basins separated by low free-energy barriers. MD simulations provide a means to explore the conformational ensemble of a native basin, and possibly transition into metastable basins at longer time scales due to larger free-energy barriers that separate basins for metastable states. Thermal fluctuations typically govern dynamic processes on millisecond or longer timescales where many functional processes take place, as shown in Figure 4. This puts a severe limitation on MD simulations to reach time scales that are functionally relevant due to computing constraints.

The practical goal of performing MD simulations is to observe molecular processes in more detail than can be obtained from experiments by sampling conformations in the form of a time series. A theoretical goal that is difficult to achieve in practice is to compute thermodynamic properties through the partition function of statistical mechanics. How much information can be extracted from time series data greatly depends on whether the process is stationary, and the sampling adequacy. The partition function requires a sum over all accessible states of a system with appropriate statistical weights. Fortunately, almost all accessible states have negligible probability. Only the most probable states need to be explored in practice. Furthermore, by obtaining probability distributions based on counting from sampling, it is possible to obtain free energy without directly calculating a partition function. While these simplifications tremendously help mitigate the problem of under-sampling, barriers in the FES greatly hinder MD from exploring the functionally relevant conformational space. In general, accuracy increases as more conformations are sampled from different free-energy basins. To increase the ability for MD to sample a wider range of conformations, it is necessary to consider enhanced sampling methods.

#### 3.2.1. Enhanced Sampling Methods

A few approaches are available to increase sampling to capture the relevant conformations for protein function. The oldest approach is to use coarse-grained models to simulate the motions of proteins on longer time scales [101,102,103]. All coarse-grained models sacrifice some information from the system to speed up calculations, achieving more sampling for the same amount of computation time. It is worth noting that there are other types of models that are not along the lines of MD simulation that model protein dynamics and/or conformational ensembles. The elastic network model (ENM) [104] and the DCM [37,38] are two very different examples of a coarse-grained model. A coarse-grained model commonly used today for MD simulation is the Martini force field [105,106], which replaces certain groups of about 4 atoms in amino acids by single elements called beads and applies physics-based effective potentials. Another choice is to use a forcefield based on the potential of mean force (PMF), such as the United Residue (UNRES) [107,108] model. This model groups atoms by their interactions, such as dipole interactions and electrostatics. Coarse-grained forcefields remove information about a structure it simulates; however, through backmapping [109], approximations to the original all-atom coordinates can be obtained. Several approaches have integrated deep learning into the prediction of coarse-grained forces, including DNNs [110,111], GNNs [112], and ensemble-based gradient boosting methods [113].

Replica exchange [114,115] is a physics-based method that involves performing many MD simulations in parallel at different temperatures. At a periodic time interval, the atomic coordinates are randomly swapped between systems and momenta scaled appropriately, which forces the conformations observed at different conditions to re-equilibriate. Higher-temperature systems will be able to move between FES minima more easily. One can also use the same idea with other variables, such as a collective variable or even forcefield parameters [115,116], as seen in the method of Hamiltonian Replica Exchange or (Replica Exchange Umbrella Sampling).

Simulated annealing [117] can be used to generate conformations during the exploration of the FES, which is controlled by starting at a fictive high temperature and cooling slowly. However, in this method, many such runs are repeated so that different basins are explored. Steps are taken stochastically using Monte Carlo methods that always move to lower-energy states when found, but have some probability to move to a higher-energy state during the exploration process. At high temperatures, the system can take large steps to explore the FES, then, as the temperature decreases at a constant rate, the system explores the full shape of the FES. This method can be used, for example, in folding simulations to find the lowest energy conformation. When performed in a mass parallel scheme, such as a swarm approach, a set of lowest energy conformers can be found by aggregating results. This simulated annealing-driven MD approach enables the native protein structure (or a set of representatives for non-native states) to be determined [118].

Lastly, non-equilibrium MD combined with Monte-Carlo methods are used to enhance conformational sampling. This approach can be classified into two schemes: introducing biases based on collective variables that are predefined, or biases placed on the Hamiltonian of the system with a potential energy that fills in the FES the longer the system spends time in a basin. Both methods increase in bias to push the system out of a basin the longer it resides there. In this way, more basins can be explored much faster than in physical reality. The objective is to capture a diverse set of conformations, not to provide real-time dynamics. These methods are often referred to as metadynamics [119,120].

In the first class of methods, a biasing force estimates the free energy along a particular reaction coordinate by computing the average force, and computing the FES by thermodynamic integration. In the second class, one way to fill up free energy basins is to add positive potentials in the form of Gaussian functions. Another method called Adaptive Biasing Force (ABF) [121,122] uses this by invoking the concept of the PMF to try and flatten out the the potential barrier along a reaction coordinate by applying a force which counteracts the PMF. When bias forces are used to perturb the underlying potential energy, re-weighting of probabilities is required to obtain equilibrium statistics and [123].

These methods have been widely applied to MD simulation, and combinations of these classic methods with ML algorithms to further improve FES sampling have been explored extensively. In the following subsections, applications of ML to various aspects of these enhancement algorithms are reviewed.

#### 3.2.2. Identifying Collective Variables

Many enhanced sampling algorithms rely on an *a priori* defined collective variable (CV), defined using expert knowledge by hand-selecting a set of residues to track coordinates, distance pairs or dihedral angles. Recent work has incorporated empirical data for CV selection, such as using co-evolutionary residue couplings [124] or the NMR $S^{2}$ parameter as a metric for conformational entropy [125]. More generally, ML provides an automated approach to CV discovery. The benefit of non-curated collective coordinates is that an unbiased approach to coordinate discovery allows hidden information with functional relevance to be identified without investigator bias.

The natural description of protein dynamics is the 3D coordinates of each of the *N* atoms in the molecule. However, the 3N dimensional space is too large to efficiently sample functionally relevant processes. Collective coordinates should be low-dimensional representations of the complex biomolecular process that is to be sampled [126,127]. The transformation to CVs should be a differential function of the 3N atomic coordinates. A traditional approach for finding collective coordinates is to use dimension reduction via unsupervised ML such as PCA [128]. Normally, it is best to use only a select set of heavy atoms, such as carbon alpha atoms, along the backbone. PCA has been implemented directly in many MD packages, including GROMACS, as well as part of standalone MD simulation analysis software such as JEDi [129], MDAnalysis [130] or ModeTask [131]. These methods are excellent for extracting the large-scale motions from a protein.

Large-scale motions are effectively represented in a low-dimensional space and are used to accelerate MD simulation [132] and identify sampling boundaries of the FES to seed further rounds of MD simulations [133]. Non-linear approaches such as kernel methods or manifold learning provided added complexity, but removed much of the intuition for a learned CV. These methods include local linear embedding [134], isomap [135], sketch-map [136], and diffusionmap [137]. Recently it was demonstrated how classification algorithms in supervised ML, such as SVM and linear regression, can be used to define CVs to differentiate two known end states of a biochemical process [138].

A popular method for CV determination as well as analysis of MD simulation data is time-lagged Independent Component Analysis (tICA) [139]. ICA is a method of signal processing that aims to reduce a high-dimensional dataset into a small set of the most statistically independent components. The original aim of ICA was to solve the so-called “cocktail party” problem, in which a mixed signal is transformed into its component signals [140]. In tICA, the ICA problem is extended to the generalized eigenvalue problem:$$C\left(t+t_{0}\right)V=C\left(t\right)\Lambda V\;\;,$$
where *V* is a matrix whose columns represent the collective independent components, Λ is a diagonal matrix of eigenvalues, C(t) is the system covariance matrix, and $C(t+t_{0})$ represents the time-lagged covariance matrix, with $t_{0}$ being the lag time between samples being compared [141]. The time lag is chosen depending on the time-scale of the processes of interest. By solving this generalized eigenvalue problem, the independent components, *V*, that maximize autocorrelation are found. The eigenvalues, Λ, represent the autocorrelation of the data at the lag time. The time scale of the motions represented by the i-th independent component, $\tau_{i}$ is estimated by $\tau_{i}=\frac{t_{0}}{ln(\lambda_{i})}$ [139]. Usually the slowest changing processes are of interest. Because tICA is able to identify long time-scale motions, it has been used to generate CVs for metadynamics [142] and is widely used for constructing Markov models [143]. The software PyEMMA [144] and MSMBuilder [145] construct Markov models for molecular simulation using tICA, as well as other reaction coordinates.

Neural networks have also been used to parameterize CVs because they are able to be computed on the fly during the MD simulation. Moreover, neural networks are differentiable, which lends to computing biasing forces [146]. Autoencoders are neural network architectures which can be used for dimension reduction and generative modeling. A neural network called the encoder predicts a low-dimensional latent space representation of the input, and the latent space points can be used to reconstruct the input by a corresponding decoder neural network. The encoder then acts as the transformation to the CV, and the decoder provides the inverse function. Autoencoders have been used to find CVs or bias potentials for enhanced sampling of molecular systems [147]. Work has also been carried out to generalize these models to allow the latent space variables to express periodic CVs and impose a hierarchical ordering on learned CVs [148]. The ability to rank CVs occurs naturally in other algorithms such as PCA, and is generally useful if one wishes to interpret the information expressed by a CV.

The low-dimensional representations learned by autoencoder models have been used in several applications to construct a Markov state model (MSM) to characterize long-timescale motions [149,150]. Similar to tICA, a time-lagged version of the model aims to predict a sample or latent point $t_{0}$ later [151]. An ensemble of autoencoders can be used to iteratively sample and bias simulations on the fly, first trained on unbiased simulation and then used to determine conformational states that are poorly sampled. Furthermore, autoencoders can be combined with umbrella sampling to sample the FES [147].

The variation autoencoder, VAE, is used for probabilistic generative modeling. VAEs work similarly to autoencoders in that they learn an encoder and decoder model. However, rather than learning discrete points in the latent space, the model learns the parameters {μ} and {σ} of a multivariate distribution over the latent space variables *z*. This distribution is sampled to produce a point $z_{0}$, which is fed into the decoder to reproduce the input *x* during training, and subsequently generates a new sample during test time. The model is trained to minimize both reconstruction error, so that the decoder can generate new points of *x* given a latent point *z*, and the Kullback–Leibler divergence between the learned distribution in the latent space of *z* and a normal distribution. This second training term constrains the learned distribution on *z* so that it may be efficiently sampled without the need for the encoder after training. VAEs have been used in combination with other methods to find reaction coordinates [152]. As an example, preselected features using the Automatic Mutual Information Noise Omission (AMINO) [153] to train a re-weighted Autoencoded Variational Bayes model (RAVE) [154] for a study on the dissociation of drugs from GPCR.

A related method which blends ideas from VAEs and tICA has recently been described for finding slow reaction coordinates by characterizing a MSM for protein dynamics, called the Variational Approach to Markov Processes (VAMP) [155]. In VAMP a variational method attempts to directly approximate the left and right singular functions of the Koopman operator which controls the time dynamics of the system on a set of low-dimensional coordinates [156,157,158]. In this approach [159], a neural network architecture called VAMPnets was developed, where two fully connected networks parallelly predict the transformation of ${\vec{x}}_{t}$ and ${\vec{x}}_{t+\tau }$ onto the Koopman coordinates by training the network to minimize the VAMP-2 score ($R_{2}$), given by $R_{2}=\left|\right|C_{00}^{-1/2}C_{0\tau }C_{\tau \tau }^{-1/2}{\left|\right|}_{F}^{2}$. $C_{00}$ and $C_{\tau \tau }$ are the non-time-lagged covariance matrices of the Koopman coordinates at time t=0 and t=τ, $C_{0\tau }$ is the time-lagged covariance matrix, and $\left|\right|\cdot {\left|\right|}_{F}^{2}$ represents the Frobenius norm [159]. This approach has been used to explore the kinetic landscape of protein dynamics by using the learned Koopman coordinates to parameterize Markov models [160,161].

#### 3.2.3. Automated Potential Biasing

Methods such as metadynamics [162] and adaptive biasing force (ABF) rely on perturbing the dynamics of the system, such that the model is driven toward regions of low sampling. To do this, models must either compute the biasing force or the biasing potential to add to the Hamiltonian of the system. The system is often projected onto CVs before biasing forces or potentials are computed to reduce the complexity of the problem. This process is performed adaptively on the fly, constructing the bias from many shorter MD simulation runs. In metadynamics, the bias is built by successively adding small Gaussian perturbations, which accumulate in well sampled areas of the FES, so that the bias potential effectively shadows the underlying potential energy surface. ABF attempts to lower barriers between free-energy basins by matching the estimated PMF. Both methods rely on sufficient sampling of the underlying potential energy surface before updates to the adaptive bias can be made.

Here, we highlight three examples of how neural networks can be used to compute adaptive bias potentials. The first method, called NN2B [163], is similar to metadynamics. However, NN2B replaces the computation of a biasing potential to add with a high-dimensional density estimator that gets smoothed over configuration space by an ANN. While the biasing potential cannot be computed in real time, the neural network and density estimation allows the model to handle higher dimensional representations. Another model, Force-biasing Using Neural Networks (FUNN) [164] trains a neural network to directly predict biasing forces on a system, rather than potential energy, at a given CV value based on Bayesian regularized neural networks [165]. The disadvantage of this method is that each bin in CV space must be sampled to train the network. However, once the network is trained, it can be used to predict the mean force for any value of the CV, even if it has not been previously sampled. The third method for enhanced sampling is Deep Enhanced Sampling of Proteins (DESP), which uses a VAE-based model [166].

#### 3.2.4. Clustering Conformations and Markov Model State Space Partitioning

Clustering data is a task that is ubiquitously used in data science and within ML methods. This section will describe many clustering methods, but the main application of interest is to quantify the similarity between conformations. Clustering conformations is an essential element of constructing a MSM. Often, a trajectory can be clustered into hundreds to thousands of microstates to describe the kinetics of biophysical processes.

Clustering is an unsupervised ML task in which a space or set of inputs is divided into groups based on how similar or different inputs are from each other [167]. At the most basic level, clustering methods can be clustered into two types of attributes: (1) what similarity/distance measures to choose and (2) what clustering algorithm to follow. Often, the number of states is left as a hyper-parameter for the researcher to optimize. Clustering of a trajectory can occur in the full coordinate space of the system, but is more commonly applied after the system is transformed into collective coordinates which characterize the progression of the biophysical process being modeled.

Clustering algorithms are sensitive to the type of distance metrics used to quantify similarity between two conformations. A common approach is to use the RMSD and TM score [168]. The RMSD distance metric compares the euclidean distance between the 3D coordinates of two structurally aligned structures [129,169,170,171]. Although commonly used due to its simplicity, RMSD is sensitive to the size of the systems being compared. The TM score is normalized on the range [0,1] to make it independent of the system size. The TM score uses a weighting scheme to normalize distances at different size scales.

An ongoing issue in comparing molecular structures is the requirement of a consensus set of residues to facilitate the comparison. After the residues are selected, the conformations to be compared must first be structurally aligned into the same frame of reference. In general, as the consensus set of residues becomes large, distance comparisons become less useful because of the curse of dimensionality. This is why the coordinate space is often transformed into low-dimensional, informative representations prior to clustering. Linear distance metrics such as euclidean distance $d_{ij}=\sqrt{\left({\vec{x}}_{i}-{\vec{x}}_{j}\right)\cdot \left({\vec{x}}_{i}-{\vec{x}}_{j}\right)}$ or the Manhattan distance $d_{ij}=\Sigma \left|{\vec{x}}_{i}-{\vec{x}}_{j}\right|$ have been widely used for clustering applications. These metrics are part of the Minkowski family of distances and are natural choices for clustering vectored data such as conformational coordinates. However, this logical approach performs poorly when different features have different scales [172]. To remove the multiple-scale problem, features are normalized prior to clustering or non-linear distance metrics with saturation limits can be used. Some metrics that are commonly used in clustering are the mahalanobis distance $d_{ij}=\sqrt{{\left({\vec{x}}_{i}-{\vec{x}}_{j}\right)}^{T}\Sigma_{ij}\left({\vec{x}}_{i}-{\vec{x}}_{j}\right)}$, cosine similarity $d_{ij}={\vec{x}}_{i}\cdot {\vec{x}}_{j}/\sqrt{{\vec{x}}_{i}^{2}{\vec{x}}_{j}^{2}}$, or Pearson correlation [173].

As applied to molecular simulation, the major classes of clustering that are popular are: geometric or partitioning methods, hierarchical methods, and model-based methods including spectral clustering [167]. Geometric clustering typically divides the data into clusters by partitioning the space of points into different clusters. The k-means clustering algorithm is one of the simplest, yet most commonly used method for this application. It assigns the data to a cluster based on which of a set of *k* cluster centriods it is closest to. The centriod locations are recomputed, and this process is iterated until convergence is reached. Geometric clustering is easy to implement and has been effective with MD data, and thus has been included in analysis packages such as MSMBuilder [145].

Hierarchical clustering constructs a dendrogram of the data points by iteratively grouping data points or clusters of data points one at a time until either the root is reached, or a specified number of distinct clusters is found. Hierarchical clustering can be employed using a top-down approach which starts at the root and works its way to the leaves, or a bottom-up approach that starts with the leaves and works its way to the root. The bottom-up approach can be referred to as agglomerative clustering. An example of hierarchical methods used in clustering protein conformations is the Bayesian Agglomerative Clustering Engine (BACE) method.

BACE [174] is an agglomerative clustering method which merges microstates by computing the Bayesian likelihood factor, or BACE Bayes factor, that the micro-states belong to the same or different macrostates. This method was inspired by the hierarchical nature of protein FES, where microstates belonging to the same macrostate will be much more statistically similar to each other than microstates in another macrostate. This method addresses the problem of statistical uncertainty in PCCA and PCCA+.

It is worth noting that model-based clustering assumes an underlying model for the data, which is exploited to cluster data points. To analyze MD trajectories, spectral clustering methods are used, which decompose a particular matrix using eigen-vector/value (spectral) decomposition [175]. In particular, a set of data points is treated as nodes of a graph which are connected by edges whose weight corresponds to similarity. A graph Laplacian matrix, $L=1-D^{-1}W$ is constructed from the graph adjacency matrix *W*, converted to similarities by a transformation of the users choice, and diagonalized. The resulting matrix of the first $n_{c}$ nonzero-eigenvalue eigenvectors ($n_{c}$ being the number of clusters) per data point are used as a new vector representing the data point, and can be clustered by another method such as *k*-means. According to spectral clustering theory, the number of clusters within a dataset corresponds to the number of eigenvalues of the Laplacian matrix that are equal to 1, or practically, within a tolerance. The most commonly used spectral clustering approach is Perron-Cluster Cluster Analysis (PCCA) and Robust PCCA (PCCA+).

PCCA and PCCA+ [176,177] is a model-based spectral clustering method which is ideal for clustering states for Markov chains. It uses the transfer matrix of a Markov chain, *P*, to construct the Laplacian matrix, where in the above equation for *L*, $D^{-1}W=P$. For a perfectly uncoupled Markov chain, the matrix *P* will become block diagonal under appropriate permutations, indicating that data points within a cluster only form edges with other data points in the same cluster within the graph representation of the dataset. The eigenvalue spectrum will have a Perron Cluster of eigenvalues at 1, allowing for the spectral clustering. This method has been used to construct Markov models and analyze rare dynamic processes in MD simulation [178]. A weakness of this method is that a sufficiently large amount of MD data must be collected so that the transition matrix is well sampled.

Methods of clustering are not restricted to an established clustering method in ML. As an example, Super-Level-Set Hierarchical Clustering [179] blends multiple methodologies together as it clusters conformations using the density of states. Initially, the conformations are clustered into microstates via *k*-means and then hierarchically ordered based on the density of the state, which is proportional to the number of conformations within the state. The microstates are then divided into *n* clusters, such that the number of conformations (not microstates) are about equal between them, so that the resulting set with less microstates is more dense in conformation space. Density sets are accumulated into a super density set, such that the *i*-th super density level contains all of the density sets 0 to *i*. Spectral clustering is performed at each super density level to reveal kinetic clusters, which form successively more connected graph representations with increasing set level. This allows clustering to be performed hierarchically and reveals structured information about the underlying FES. Further extensions of this algorithm, called the Hierarchical Nyström Extension Graph, have been published by the same authors [180] to account better for the high-dimensional nature of protein dynamics.

Another approach to clustering is the Most Probable Paths algorithm [181], which takes a set of *n* microstates and converts them into more coarse-grained macrostates. The trajectory is first reduced into a low-dimensional form by PCA, then the free energy is computed and microstate transition probabilities are approximated via the assumption that $G_{i}=k_{b}Tlog\left(P_{i}\right)$, where $G_{i}$ is the free energy of state *i*, $k_{b}T$ is the Boltzmann constant times temperature, and $P_{i}$ is the probability of being in state *i*. The transition path is followed by jumping to the nearest neighbor state with the highest probability (the lowest $G_{i}$) until the path terminates or reaches a loop. All states that terminate at the same lowest free-energy state are lumped into a composite state, and the algorithm is repeated until the number of macrostates converges.

Other clustering methods are available, as previously reviewed [182] and described in much more detail. It is noteworthy to mention Renormalization Group Clustering [183], Automatic Partitioning for Multi Body Systems [184], Sapphire based clustering [185], and self-organizing maps as unsupervised neural networks [186]. It is clear from this brief survey that additional clustering algorithms that learn on MD simulation data will be developed and introduced. From our own research, we recommend a new direction toward developing conformational comparisons that are not directly tied to common sets of residues. The traditional approach of atom to atom comparison makes it difficult to compare conformational ensembles across larger sets of proteins that have large variations in sequence identity. This paradigm shift would require replacing structural alignments between corresponding atoms [129] in terms of physical properties that go beyond atomic coordinates. Such an approach would allow environmental and dynamic effects to be quantified, assist protein evolution studies and benefit ML methods for structure prediction.

### 3.3. Protein Stability

Stability is a question that is central to protein science. In the 1950s, Pauling’s elucidation of the native contacts of the protein backbone in helical structures spurred the field of structural biology [7,187,188]. This quickly evolved to the current day perspective: stability of proteins is functionally important from the mechanical, thermodynamic and kinetic point of view as depicted in Figure 5. From the results of XRC, the common view among biochemists and molecular biologists is that functional proteins are stable with a well-defined, mostly rigid structure, and some flexible loop regions protruding into solvent. This corresponds to the case when a protein maintains a tight conformational ensemble at the bottom of a free-energy funnel, consistent with an enthalpic dominance in the free energy of the protein.

Empirical characterization methods for protein structure and stability are biased one way or another. The biases originate from what is measured. For example, in crystallography, it is only possible to observe mechanically stable proteins, because they pack tightly when placed in an environment with significant concentration in the solution state. In contrast, NMR measures protein dynamics in solvent, and includes information about the solvent. For example, it is possible to observe wakes in solvent paramagnetic relaxation enhancement NMR [189]. The merging of this heterogeneous empirical data over the last half century has transformed fundamental physics concerns into data science. It is now important to develop methods and concepts that work with conformational ensembles through their probability distributions and describe how different environmental conditions modulate conformational ensembles, ushering in big data analysis.

Since the operative word in data science is “data”, it is worth describing where data for protein function come from. Databases such as SCOP and CATH have been valuable for learning how to perform secondary structure prediction and early homology modeling [9,190,191,192]. Historically, the accession of structural and sequence-based data spawned regression and statistical methods to obtain predictive models [191,193,194]. In a similar spirit, other empirically derived data yield metrics such as the gravy score among many other sequence propensity scales [195]. Notice that the depositories concern themselves with data that define a particular property that is not directly related to function. The protein function analysis part of the problem comes in because these data are clustered based on categorical labels that specify which protein is functional or not, making data mining plus clustering a standard bioinformatics approach. Although this data mining approach has been extraordinarily useful, methods that rely on data collections are subject to hypothesis formation bias [196].

An important class of proteins are Intrinsically Disordered Proteins (IDPs). An IDP has disordered regions, which can mix with ordered domains or small well-defined structural motifs. Often IDPs undergo induced conformation change upon binding [197]. Disorder-based functionality complements the known function of ordered proteins and domains [198]. The prominent characteristic of IDPs is that these biologically active proteins/domains do not coalesce into stable 3D structures under physiological conditions without some inter-molecular interaction. Naturally, structural data for IDPs or intrinsically disordered regions are lacking. As such, the contrast between globular proteins and IDPs highlights how training data can lead to hypothesis formation bias due to sampling bias inherent in a structural database. When ML methods are trained on structural and other empirical data, potential sources for bias hypothesis formation should be considered.

Working with conformational ensembles provides a direct route to elucidate protein stability, dynamics and function. However, there remain open questions about the best way to generate, represent and quantify conformational ensembles. Early work includes taking a statistical approach [199], knowledge-based analysis [10] and unsupervised clustering [200,201]. Eventually, neural networks were applied to recognize protein folds [202] or to self-improve a predicted folded structure [203]. Subsequently, predicting optimal sidechain packing was addressed in pytorch implementations [204,205]. Using specialized transformations, such as Voronoi tessellations, and CNN has been shown to yield state of the art fold modeling [206]. In addition, contact map predictions have been performed using various methods utilizing aggregated sequence and statistical data [207], which in part was leveraged by alphafold, where a DNN is applied to contact maps and can be leveraged to predict multiple conformations [84,208].

Predicting stable conformations using ML has been achieved using neural networks [209] by operating over local conformation changes generated through physics-based minimization. In the context of thermodynamic and kinetic stability, ML has added a new twist into protein fold prediction. To simplify the analysis as much as possible, a large body of work has focused on fast folders or ultra-fast folders [210] as model systems. Utilizing Bayesian decision trees in tandem with MD has been shown to properly identify folding pathways [211]. To reduce the computational costs of protein folding MD simulations, ML-guided dynamic frameworks have been introduced with some success [212].

Typically, for initial descriptors, these models use a combination of structural features, sequence information using singleton or multiple sequence alignment, and structural topology representations such as contact or distance matrices. Feature crafting is a key element of well-designed learning algorithms. In addition to sequence propensity scores, structural features include atomic packing [213], electrostatics [214], meta dynamics from MD, and structural families, such as those found in SCOPe [10]. These features in combination with various software packages for parsing and curating structural elements with adaptable objects suitable for ML, such as pytorch objects, will likely pave the way for the next generation of structural bioinformatics [215].

There is a resurgent interest in predicting protein stability, mainly from sequence information, and possibly from static structures. For example, emergent sequence features from transformer processes with alternating self-attention with feed-forward neural network connections [216] have been applied to embed sequence spaces. Along these lines, with byte pair encoding compression and training on ΔΔG data, a mutational predictor for structural stability directly from a sequence was created [217]. Furthermore, a self-attention-based variant of a GAN has been applied to learn the sequence diversity to generate new functional protein sequences [218]. To obtain improved predictions for stability, solubility and binding affinity, recent works integrated 3D information into sequence embeddings using a message-passing neural network consisting of encoder-decoder layers trained on atom distances [219]. Possibly, self-learning will take into account thermodynamic attributes such as enthalpy and entropy, of stable folds of proteins, along with folding pathways and functional classifications in tandem [220].

In our view, a purely sequential approach will fail to give robust protein stability predictions. The environment dictates the nature of the conformational ensemble in conjunction with the constraints placed by a sequence. Without context, stability is undetermined. Specifically, solvent and conformational entropy effects cannot be neglected when quantifying protein stability. Therefore, the ML methods that incorporate environmental and conformational ensemble information will likely perform well in the long run. In the short term, good results may be obtained by working with small datasets that have systematic trends; however, experimental evidence in the late 1980s and early 1990s [35,36] suggest to us that the same mistakes will be repeated under the guise of non-linear black-box magic.

#### 3.3.1. Role of Environment

Protein stability highly depends on environmental factors. Most work takes into account how protein stability can be altered with different formulations or how stability can be shifted by small molecular ligands. While numerous factors are present in the crowded cell, we limit our review to the description of protein stability as a function of the chemical composition of small solute molecules. This simpler situation is already a rich and important problem, germane to synthesis and purification of proteins in practice.

Practical computational approaches employ datasets that characterize protein sequence/structure characteristics and correlate this information to solubility using ML models. A graph-convolutional network trained on soluble protein datasets, such as eSOL, treats solubility prediction as a regression problem. By including contact maps along with a variety of sequence and structural information about the protein, predicted accuracy benchmarks had an R^2^ of 0.48 and AUC above 0.8 [221].

In another line of research, improving solubility by modifying moieties through solubility tags is of great interest. An algorithm that creates these for peptides was found to increase solubility in empirical studies by over 100%. This work also trained on the eSOL database using support vector regression to inform a genetic algorithm to optimize the tag additions for the given sequence [222]. Similar work utilizing NLP techniques learned good soluble fragments from the TargetDB [223]. Another approach combined dilated CNN layers with residual ANN and a Squeeze-and-Excitation layer in effort to extrapolate long-range relationships along a sequence to relate to measured solubility [224].

The ability to increase protein stability without loss in protein function efficacy is the objective of designing protein formulations. A key lacking aspect seems to be understanding this for a specified aqueous co-solutes, whether ionic or aliphatic in nature. One cannot separate the conformational ensemble of a protein in an aqueous solution without taking into account the exchange of protons from the protein to the medium. From explicit MD simulations of constant pH, training a convolutional neural network in tandem with dense layers for pKa prediction was shown to be fairly robust for soluble proteins by using MD simulations to train ML models [225]. A model for force computation called high-dimensional neural network potentials have been shown to handle atomic electrostatics with good accuracy with moderate computation times [226]. Going forward, it should be possible to include details of electrostatics in such an evaluation in order to create correlations between protonation states based on pKa in aqueous solutions to the electrostatic potential of the protein.

The environment of a protein need not be of aqueous nature. There has been recent work with ML to predict membrane interfaces of proteins using reinforcement learning on structural data generated from MD, after determining appropriate CVs. Utilizing an ensemble of predictors tied together with a meta classifier, exhaustively tested and optimized, yielded state-of-the-art accuracy under limited conditions [227]. In a similar fashion of training, MD-based training data to generate CVs that are tractable have also been used to understand aggregation of larger proteins, such as antibodies, where the environmental question is defined by the protein target of interest [228,229]. Post-translational modification (PTM) sites have also been tackled, such as glycosylation [230] using RF algorithms to predict sites from sequence input or phosphorylation sites in a similar fashion [231].

In each of the cases discussed above, a key dataset is used to train a model for inferring relationships between sequence/structural information and how the protein will react to the environment. In each of these applications, extrinsic information is being inferred on data from a secondary source. A key opportunity is to look for transferability in the ML model to account for changes in the environment.

#### 3.3.2. Protein Engineering Through Mutagenesis

Mutating residues to modify protein function or achieve a specific molecular property is the earliest examples of protein design, also known as re-design and protein engineering. We are interested in the computational biology underpinnings of in silico structural based platforms that proceed along these lines [232,233,234,235,236]. Historically, plasmid insertions gave biochemists the ability to deduce functional sites and facilitate molecular engineering, and these procedures eventually led to modern adaptions [237,238]. Combinatorial approaches, threading along backbones and more detailed knowledge-based methods have dominated the field [69,191,239]. Fixed backbone designs lacked predictive power for binding events due to the necessity of understanding dynamics or flexibility of the protein [240]. Early attempts to solve this problem applied dimension reduction techniques or SVM [193,241].

To help quantify the shifting of stability in a protein, ProTherm was created as an empirical mutational database that includes the attributes of ΔΔG and change in melting temperature. Application of ML models to the curated ProTherm dataset generated mutational landscapes utilizing SVM and RF decision regression. These approaches obtained fair predictive power for the dataset, when compared to naive Bayes classifier, K nearest neighbor, partial least squares, and an ANN [242,243]. A deep neural network predictor of ΔΔG using this same training dataset outputs ΔΔG values for mutated sequences, as determined by multiple sequence alignments [244]. When this model was validated using the same data source, linear correlations in the range from 0.6 to 0.7 were obtained. However, when tested against novel sequences, the correlation ranged from 0.7 down to 0.1, with predictions sensitive to specific attributes of the protein function. Optimizing protein interfaces through mutagenesis has been successfully achieved using a RF approach [245], which compared favorably to other mutagenesis predictors such as SKEMPI. Most results to date use training datasets that over-represent specific protein families, which leads to poor translation to other families. It has been noted that generalizing a mutational stability model must be undertaken with great care for each specific problem addressed [196].

Most recent advancements in predicting stability changes of a protein due to mutation include hallucinations via diffusion models applied to protein structures to generate novel composite single and multi-domain globular proteins [246]. Improving interactions and stability of structural space from their relationship to sequence space was deployed using a message-passing neural network with empirically deduced success [219]. Methods that minimize the number of mutations to shift stability are available [247]. In addition to addressing the solubility problem, TopologyNET a DCNN also predicts stability changes upon mutagenesis using a neural network design, demonstrating that the two problems are often entwined [248]. Pipelines are being developed that piece together ML packages with evaluation methods such as alphafold, Rosetta, or MD simulation evaluation to find possible realistic folding proteins [249].

In most works to date, the process of prediction followed by refinement using known features is applied. Prediction of stability shifts due to mutagenesis requires structural and thermodynamic considerations, which makes it challenging to achieve reliability.

#### 3.3.3. Protein–Protein Interactions

Understanding molecular function extends past the intrinsic characteristics of individual structures/sequences and involves the dynamics of protein–protein interactions (PPIs). These interactions can occur in many ways, such as permanent (i.e., irreversible binding) or transient relationships (i.e., intracellular signaling interactions) [250]. Predicting if and how proteins will interact is an ongoing computational/experimental challenge that is driven by a large number of confounding factors. In vivo, a protomer’s localization, concentration, and local environment can affect the interactions between proteins, which are vital to control the composition and oligomeric state of protein complexes [13]. The majority of PPIs cannot be neatly categorized into a dichotomy of obligate or non-obligate interactions; there exists a gradient between the two. Furthermore, the stability of complexes is heavily dependent on both the physiological conditions and environment [13,17]. The ML methods aimed at predicting PPIs can be broadly categorized into two forms: structure and sequence-based approaches; additionally, there are various objectives for PPI prediction. These objectives include: the binary task as to whether two proteins generally interact (class 1) or not (class 2); PPI site prediction; and PPI binding affinity prediction. In this section, the focus will be on the general problem posed as to whether two given structures interact or not.

Sequence-based PPI prediction methods generally begin with two protein sequences and result in a score which ranks the probability that an interaction occurs. Encoding the complex information required for robust PPI predictions is of vital importance while performing feature engineering [251]. Furthermore, an issue arises due to the variable length of proteins under PPI analysis. Many ML methods require input feature vectors of equal length; this often involves the padding/truncation/aggregation of sequences [252], where the inevitable loss or distortion of information will occur. A non-exhaustive list of ML methods utilized for sequence-based PPI prediction is presented in Table 3. Structure-based PPI predictors are not constrained by the same problems as sequence-based approaches, such as the loss of important predictive information during feature construction. A list of structure-based PPI prediction models from 2018 is also presented in Table 3.

The majority of ML methods for PPI prediction are sequence based (static structures), and do not account for conformational ensembles or the role played in PPIs by local environment. Despite the use of high-throughput techniques in discovering PPIs, the coverage of experimentally determined PPI data remains poor [253]. This translates to a lack of labeled data for ML model training, which directly affects the generalizability of predictions on previously unseen systems. Given the delicate nature of constructing the correct features for sequence-based PPI prediction, it is a natural extension to utilize neural networks that have the capability of identifying useful latent features simultaneously during model training. This trend can be observed from Table 3 as the successful PPI prediction ML models have largely shifted into deep learning architectures.

**Table 3 biomolecules-12-01246-t003:** Machine learning methods for PPI prediction from 2018 to 2022.

Paradigm	Method	Year	Model ${}^{1}$	Stand Alone ${}^{2}$	Webserver ${}^{3}$
Sequence	DPPI [254]	2018	CNN	Yes	No
	EnsDNN [255]	2019	DNN	Yes	No
	MDPN [256]	2019	DPN	No	No
	CNN-FSRF [257]	2019	CNN + RF	No	No
	S-VGAE [258]	2020	GCNN + VAE	Yes	No
	EnAmDNN [259]	2020	DNN + Att	Yes	No
	PCPIP [260]	2021	SVM	No	Yes
	CAMP [261]	2021	CNN	Yes	No
	Balogh et al. [262]	2022	GAN	Yes	No
	TAGPPI [263]	2022	GCNN	Yes	No
Structure	Daberdaku et al. [264]	2018	SVM	Yes	No
	BIPSPI-structure [265]	2018	DT	Yes	Yes
	IntPred [266]	2020	RF	No	No
	MaSIF [267]	2021	ANN	Yes	No
	GraphPPIS [268]	2021	GCNN	Yes	Yes

^1^ The nomenclature for ML models utilized: “SVM” (support vector machine); “CNN” (convolutional neural network); “DNN” (deep neural network); “MDPN” (multimodal deep polynomial network); “RF” (random forest); “GCNN” (graph convolutional neural network); “VAE” (variational autoencoder); “Att” (attention mechanism); “SVM” (support vector machine); “GAN” (generative adversarial network); “GNN” (graph neural network); “DT” (decision trees); “ANN” (artificial neural network). ^2^ This column indicates whether a standalone program is available (“Yes” or “No”). The hyperlink for “Yes” redirects to the corresponding repository. ^3^ This column indicates whether a webserver is available to users (“Yes” or “No”). The hyperlink for “Yes” redirects to the corresponding webserver URL.

#### 3.3.4. Role of Rigidity in Disordered Proteins

Beyond the classical structure–function paradigm, which is grounded in a general lock-and-key model driven by the assumption of unique structures, is a continued interest in the role of IDPs and protein hybrids. Protein hybrids are comprised of intrinsically disordered protein regions (IDPRs) and ordered domains. Disorder-based functionality complements the known function of these ordered proteins and domains [198]. The prominent characteristic of IDPs and IDPRs is that these biologically active proteins/domains do not coalesce into stable 3D structures under physiological conditions. Consequently, they lack the structural rigidity that is often cited as necessary for function in the structure–function paradigm [269,270]. From a physics perspective, an important characteristic of IDP/IDPRs is that they exhibit a flat Gibbs free-energy landscape, not favoring one molecular conformation over another [271]. The IDP/IDPR systems exist as highly dynamic structural ensembles at the secondary and/or tertiary levels [272,273]. Understanding the conformational state space that is spanned by disordered structures is vital for a robust structure–function paradigm that will allow for high levels of recognition specificity and provide further insight into how IDPs/IDPRs interact with other proteins and molecules.

Of particular interest are the IDPRs known as molecular recognition features (MoRFs), which execute their function through a phenomenon known as disorder-to-order transitions (induced folding) [197]. While the vast majority of an IDP is flexible, these MoRFs form relatively stable structures as a rigid structural motif. Machine learning predictions of IDPs/IDPRs are especially important, as they contribute to the experimental discovery of PPIs. Previous studies have compiled/analyzed intrinsic disorder predictors [274,275]; here, a select list of predictors is provided in Table 4, which was constrained to ML methods since 2016.

The gold standard for IDP/IDPR predictor evaluation is the critical assessment of protein intrinsic disorder prediction (CAID). This is a community-based blind experiment aimed at identifying state-of-the-art IDP/IDPR predictors. In the 2021 CAID benchamrk [276], 43 methods were evaluated while using a dataset of 646 proteins curated by DisProt [277]. To evaluate the methods across different applications, three forms of the base dataset were utilized: (1) fully disordered proteins (IDPs) from Disprot; (2) fully disordered proteins (IDPs) from PDB; and (3) a binding challenge dataset where positive labels correspond to residues annotated as intrinsically disordered binding residues (IDBRs) in the DisProt database. It should be noted that the three^rd^ dataset contained 414/646 target IDPs that were absent of any positive labels that were derived from the experiment. Two overarching challenges were set; the first of these assessed performance of the predictors when the objective was the identification/prediction of fully disordered proteins (IDPs) using datasets 1 and 2. The second challenge evaluated the performance of the predictors when the objective focused on the prediction of disordered protein binding regions (DPBRs), benchmarked using dataset 3.

**Table 4 biomolecules-12-01246-t004:** Machine learning methods for IDP/IDPR prediction from 2016 to 2022.

Method	Year	Model ${}^{1}$	Stand Alone ${}^{2}$	Webserver ${}^{3}$
MoRFchibi [278]	2016	SVM	Yes	Yes
AUCpreD [279]	2016	CNF	Yes	Yes
Predict-MoRFs [280]	2016	SVM	Yes	No
SPOT-Disorder1 [281]	2017	RNN	Yes	Yes
MoRFPred-plus [282]	2018	SVM	Yes	No
OPAL+ [283]	2018	SVM	Yes	Yes
rawMSA [284]	2019	CNN + RNN	Yes	No
SPOT-Disorder2 [285]	2019	CNN + RNN	Yes	Yes
ODiNPred [286]	2020	SNN	No	Yes
IDP-Seq2Seq [287]	2020	RNN + Att	No	Yes
flDPnn [288]	2021	DNN + RF	Yes	Yes
flDPlr [288]	2021	RF	No	No
RFPR-IDP [289]	2021	CNN + RNN	No	Yes
Metapredict [290]	2021	RNN	Yes	No
DeepDISOBind [291]	2022	MTNN	No	Yes
MoRF-FUNCpred [292]	2022	BR + SVM + RF	Yes	No
DisoMine [293]	2022	RNN	No	Yes

^1^ The nomenclature for ML models utilized: “SVM” (support vector machine); “CNF” (convolutional neural fields); “RNN” (recurrent neural network); “CNN” (convolutional neural network); “SNN” (shallow neural network); “ANN” (artificial neural network); “RF” (random forest); “MTNN” (multi-task neural network); “BR” (binary relevance). ^2^ This column indicates whether a standalone program is available (“Yes” or “No”). The hyperlink when “Yes” redirects to the corresponding repository. ^3^ This column indicates whether a webserver is available to users (“Yes” or “No”). The hyperlink for “Yes” redirects to the corresponding webserver URL.

Here, we provide a brief summary for the performance of the participants and discuss general trends. The primary metric for performance during CAID is the maximum $F_{1}$ score, $F_{max}$, or the maximum harmonic mean between precision and recall over all thresholds. This is a robust metric for two reasons: (i) $F_{max}$ takes into account all predictions across the full sensitivity spectrum; and (ii) $F_{max}$ is invariant to imbalanced datasets. The largest general trend from the assessment was that the best methods utilize deep learning neural network architectures which outperform physiochemical-based methods. For the first challenge over all performance metrics (beyond $F_{max}$), the predictors fIDPnn, SPOT-Disorder2, rawMSA, and AUCpreD consistently perform in the top five, respectively, attaining an $F_{max}$ of 0.48, 0.47, 0.45, and 0.44 on dataset 1. For dataset 2 where all known structured regions are filtered out, these numbers substantially increase for the same methods to an $F_{max}$: 0.71 (fIDPnn), 0.79 (SPOT-Disorder2), 0.75 (RAWmsa), and 0.77 (AUCpreD). This exemplifies that the removal of known structured regions heavily simplifies the prediction problem. For the second challenge (DPBR prediction), the assessment across all methods shows a substantial reduction in predictive performance, indicating a need for advancement. The $F_{max}$ for the top 5 methods here was 0.214 ± 0.0134.

Given the recent non-incremental increase in the structure prediction field by AlphaFold2, it is pertinent to address recent concerns relating to AlphaFold2 and intrinsic disorder prediction. It has been noted that the disorder predictor one constructs from a predicted AlphaFold2 structure determines accuracy [294]. This is due to non-trivial underlying assumptions such as annotating residues from helices, strands, and H-bond stabilized turns as ordered while the remaining are unordered. This ultimately leads to a dramatic overestimation of disorder [274,275]. We suspect that structure prediction methods will soon embrace quantifying conformational ensemble diversity to allow for more representative predictions for disordered regions. However, this remains an open problem.

### 3.4. Protein Dynamics

Protein motion, of which there are many types, as schematically shown in Figure 6, is critical to protein function. Identifying flexible regions in proteins can be empirically achieved in several ways. The two most common methods are XRC and NMR. In XRC, the temperature factor (or B-factor) estimates the fluctuations of atoms due to thermal vibrations in their equilibrium positions from the attenuation of scattering [295]. The B-Factor has been moderately found to correlate with structural flexibility, particularly when using just carbon alpha B-factors [296], but it is not a direct measurement of flexibility.

This is because XRC only provides static structures. Crystal packing biases the B-factors during the process of solving the inverse problem for structure from the raw X-ray diffraction data, as resolution errors are folded into the B-factors [305]. In the solution, NMR works with conformational ensembles, which allows flexibility to be sampled. For example, the root mean squared fluctuation (RMSF) is moderately correlated with the $S^{2}$ order parameter [306]. There are other modes of NMR operation, and in general, NMR measures dynamics on a limited range of timescales.

In addition to these incompleteness and indirect measurement problems, directly resolving dynamical correlations among flexible regions in a protein remains an experimental challenge. Therefore, MD simulations are typically used to identify flexible regions within a protein and monitor time correlations. However, MD simulations cannot reach long time scales, and there are insufficient statistics at the longest timescales that can be reached. Furthermore, it is a challenge to identify functionally relevant dynamics both experimentally and computationally. This is because the observation of atomic motions does not answer how important these motions are for a mechanism of action. In general, non-functional motions will hide functionally relevant small-amplitude motions, especially when non-functional motions have large amplitude. Therefore, functional dynamics are likely to be missed by methods that intrinsically extract large structural fluctuations in a protein, such as PCA. Since PCA is used as a dimension reduction method, it is possible to lose important information about functional dynamics before more sophisticated ML methods are applied whenever PCA is indiscriminately applied.

#### 3.4.1. Protein Flexibility and Conformational Dynamics

For brevity, we only discuss globular proteins and IDP as two classes of proteins. Globular proteins tend to hold a well-conserved 3D shape, although parts of the structure can flex. The most flexible parts of a globular protein are located in its loop regions. In general, large-scale motions and localized flexible regions are important for protein function. Flexibility in a particular macrostate (e.g., the native state or a metastable macrostate) exists as a hierarchy in the size of certain structural elements and in terms of local minimums in the free-energy funnel. As an example of this, a full domain of a protein may undergo a concerted, global conformational change, while regions of secondary structure remain more rigid due to stabilizing hydrogen bonding. Loops, particularly those protruding into the solvent can be quite flexible in contrast with the the stable core of a molecule [39,307] Finally, at the highest resolution, residue side chains can be flexible in that they switch between different rotamer states. Flexibility at each of these hierarchical levels can help the protein perform a specific function. When comparing two or more proteins in terms of flexibility, root mean square deviation (RMSD) and root mean square fluctuation (RMSF) provide simple metrics that inform differences in global and per-residue flexibility given a ensemble of structures from, e.g., MD simulation. Additional measures are the flexibility index, which quantifies how flexibility is distributed throughout the protein, which is directly dependent on thermodynamic stability [39,308].

One approach for ML to directly predict flexibility in proteins is to use the structure data collected in the PDB to predict the experimental flexibility metrics, such as the B-factor and $S^{2}$ order parameter. The Gaussian Network Model (GNM) and the more general Elastic Network Model (ENM) [309] allow RMSF to be calculated, and this quantity moderately correlates with B-factors and $S^{2}$ order parameters [39]. Another metric that can be learned by training these networks is the Flexibility–Rigidity Index (FRI) [310,311] which can be computed for each atom (graph node) as a weighted sum of its connections. The sum of all FRI node values provides a global FRI value, which is inversely proportional to flexibility. FRI can be used as a predictor of B-factors and general stability in proteins. More recent work [312] has pitted these network models directly against ML algorithms (RF, gradient boosting trees, and CNNs) and found that CNNs provided the best prediction. Similar work to predict $S^{2}$ has used shallow neural networks [313] to predict the parameter directly from feature embeddings of 3D structures. Some more recent work [314] has been carried out using ML to validate structure distributions from CryoEM, a method gaining considerable traction in structure determination. The method uses a Gaussian Mixture Model, inspired by the structure of VAEs to approximate the electron density map, with the limits of the density corresponding to the probable area for an atom to exist.

In the literature, there are many types of ENMs [309]. In general, an ENM makes a direct assumption that the ensemble of conformations can be described as quasi-harmonic vibrations with respect to a rigid reference structure. This is modeled by sampling the bottom of a harmonic potential well, and modeling the protein as a set of masses connected by springs. The equation of motion for *n* masses is $M\ddot{\vec{x}}=-K\vec{x}$. In this model, $\vec{x}$ and $\ddot{\vec{x}}$ are the positions and acceleration of each atom or mass point, *M* is a diagonal matrix containing the mass of each mass point, and the elements of matrix *K*, $k_{ij}$ are the effective spring constant connecting points *i* and *j*. *K* is called the Hessian matrix for the model, where harmonic potentials connect to mass points. The eigenvectors represent the normal modes of motion for the system, and their eigenvalues are the corresponding frequency of that motion. The mass of each mass point will in general be different because the constituent atoms associated with each mass point are heterogeneous. However, often the masses of all mass points are set equal. These spring connections are heuristic, and the ENM is not the same as a true normal mode analysis since the ENM is set at a coarse grained level description.

In principle, equilibrium MD simulation provides the best way to investigate flexibility by directly sampling the conformational space. However, as discussed above, in practice there is usually an unobtainable computational cost, and to make matters worse, one faces massive amounts of data, which must be analyzed to quantify flexibility through dimension reduction. Nevertheless, the most popular method for extracting flexible motions from equilibrium simulations is Essential Dynamics (ED). Essential Dynamics [128] applies PCA to the ensemble of mean centered conformations by diagonalizing the covariance matrix. The eigenvectors and eigenvalues represent global motions, which can be ranked by size scale using the eigenvalues. The eigenvalues are the variance in the global motion represented by the eigenvector, where larger eigenvalues represent greater amplitude vibrations. ED can be performed on the Cartesian coordinates of an ensemble of structures, or internal coordinates such as distance pairs or dihedral angles.

A link was found between ED and ENM, where the Hessian matrix can be approximated for a sampling of conformations by the inverse of their covariance matrix, which is the matrix used in PCA. Thus, within the qausi-Harmonic approximation, the most variant PCA mode (with the largest eigenvalue) is equivalent to the lowest-frequency ENM mode (with the lowest eigenvalue) [34]. It is important to note that the limitation of both approaches is that they require the conformational ensemble to remain close to a single representative structure. This means jumping between different free-energy basins cannot be accurately described. However, at the coarse-grained level, jumping between basins is modeled as harmonic on long-time scales. In this case, these methods describe large-scale rearrangement rather than intrinsic flexibility. Depending on the application, this can be seen as a limitation or benefit of using these methods.

Some ML methods have been used to predict flexible and dynamic regions in proteins. Much of this research is driven by better docking algorithms that can accommodate flexibility in the receptor and ligand. For example, SVMs have been used as binary classification algorithms to determine whether loops have high mobility or not. Early classification algorithms often use curated features such as solvent-accessible surface areas, B-factors, and other structural quantities, which improve but likely bias predictions. Informative features for describing protein flexibility are required to elucidate which atomic motions correlate with functional characteristics. Random forest is commonly used for classification and regression tasks, but metrics such as Gini impurity [315] allow the model to understand feature importance, which provides context into how the model relates its final prediction to the input features. Exemplifying how Gini impurity can provide insight to how RF models make predictions, RF models were trained to classify conformations in beta lactamase at various stages of ligand catalysis, and using the Gini index revealed residue pairs and structural regions most important for the classification [111]. The success of using dynamics-based models to inform predictions relating to protein function reveals the deep connection between dynamics and function. ANNs have also been used to identify mobile regions of proteins. One such method to identify flexible residues in proteins is NEAT-FLEX [316]. Importantly, neural evolution is applied using a genetic algorithm to augment typologies for learning the optimal topology in the neural network to predict molecular properties.

Connecting conformational dynamics to function can be built into models that directly work with trajectories that describe protein dynamics. The tICA method, which we discussed in Section 3.2.2, does this by using time lag to let the model focus on dynamics at a particular time scale. This approach has also been applied to t-Distributed Stochastic Neighbor Embedding (t-SNE) [317], a non-linear dimensionality reduction method for efficiently reducing MD trajectories into a low dimensional space. A different approach from the Jacobs lab is to apply discriminant analysis to extract functional dynamics using a Supervised Projection Learning for Orthogonal Completeness (SPLOC) algorithm [318]. This method uses data-driven optimization to learn spatial-scale and temporal-scale independent dynamic features, which best distinguish two classes of molecules which function differently. The target for learning is fully defined by the data presented, rather than underlying assumptions [319] such as variance in PCA. The method was demonstrated by describing the dynamic differences in multiple TEM beta-lactamase, which explain differences in enzyme efficiency among several ligands [302].

Generative models provide an interesting approach to studying the flexibility of proteins, which requires training on known conformational ensembles. This data can be obtained from MD simulation trajectories. The MD data provides the model with a baseline for how the atoms move. Then the model learns low dimensional distributions that are used to generate more conformations to help fill out the low sampled regions. This provides a means for studying molecular motions in the native state. A simple, but effective method for doing this is to determine the normal modes of a protein with an ENM, and then use the modes themselves as displacement vectors for perturbing the structure [320]. This strategy was implemented for an application to investigate the opening of cryptic pockets in various proteins.

Autoencoders have also been used to generate realistic conformational ensembles [321]. A disadvantage of using traditional autoencoders is that the information is not efficiently and continuously mapped to the latent space which prohibits effective sampling [322]. Again, turning to VAE allows the model to learn continuous dynamic processes such as folding or conformation change [212]. These latent spaces can be used to study the dynamics of proteins in great detail, as shown in this study [323] that used them to learn important conformational states of GCase. Another generative method, similar to an autoencoder, but which learns the distributions in a Gaussian kernel space, is the Flexible Backbone Learning by Gaussian Process [324] (Flex-BaL-GP), which uses redundant entries in the PDB to describe alternate backbone conformations [325]. The database Conformation Diversity in the Native State (CoDNaS) is a resource for training ML algorithms to detect flexible regions in proteins.

In the early days of structural bioinformatics and computational biology applied to understanding the structure and dynamics of proteins, methods were developed to predict the most flexible regions in a protein. As reviewed above, this is not a difficult task, as there are many methods (based on ML and/or physics-based models) available to identify flexibility or mobility. It is seen that a wide range of methods yield results that are consistent with experiments that measure mobility. The more interesting question that goes to the heart of protein function is how flexible regions and their motions are correlated. The construction of conformational ensembles using MD simulation can, in principle, uncover this correlation, but to reach the timescale of functional dynamics could require calculation times that are not feasible. Our view is that physics-based modeling will be the rate-limiting step in characterizing functional dynamics, and successful models must include the effects of thermodynamic and mechanical stability and their interrelationships.

#### 3.4.2. Dynamic Allostery

Allosteric signaling is a long-range communication present in proteins that affects function, conformation stability, or interaction propensities between partner molecules [326]. Note that long range refers to distance in a 3D structure. There are many types of allostery, and there are precise definitions of the phenomena in terms of binding curve shifts [327]. A classic example of allostery is exhibited in the function of hemoglobin, where binding one diatomic oxygen ligand acts as a homotropic allosteric regulator, resulting in a lowered barrier for the binding of subsequent diatomic oxygen ligands [328,329]. Most models of allostery, including obligate and conformational allostery, are based on identifying correlated motions in sets of conformational ensembles. This review is concerned only with dynamic allostery because it has a universal mechanism that is potentially present in all proteins [330,331]. This universal character occurs because in dynamic allostery, the mechanism is driven by modes of vibration in the protein, or alternatively, how rigidity propagates through a protein structure [332].

Several normal-mode-based allosteric site prediction software exist [333,334,335,336,337]. There has been a wealth of studies that benchmark the accuracy of these models/software. It has been shown that dynamic allostery will be present for structured proteins with flexibility in limited regions [338]. Off target affectors create coupled dynamics through vibrations, which potentially alter preferential binding propensity [339]. In addition, if a protein is mutated, its sensitivity in vibrational characteristics is likely to change. As such, like other functional mechanisms, dynamic allostery is affected strongly by evolution as noted in beta-lactamase [302]. It is computationally feasible for methods detecting dynamic allostery to be extended to design proteins with planned allostery communication signals.

Attempts at classifying allosteric signaling using local features generated from MD have been attempted. Results using naive Baysian inference and SVM yielded poor allosteric predictions [340]. Recently, ML applied to MD for allosteric signaling has been performed, with a larger emphasis on the MD than ML in most applications. This is because there is a need to obtain well-sampled data to characterize the ensemble that describes the phenomenon. Good accuracy was reported in one work using SVM and random forest on MD-docking data with known categorical values for drugs [341]. Deep neural networks have been applied to short MD trajectories using a specialized metadynamics routine, called neural relational inference MD. In this model, an allosteric signal is put through a VAE to interpolate conformations and determine the communication pathways within the protein [342]. The combination of extreme gradient boosting (XGBoost) and GCNNs for allosteric site prediction was found to predict conformational allosteric sites on static structures tested, without a need for heavy simulation computations after being trained on known allosteric proteins to learn topological connections that define the phenomenon [343].

Some limited work involving engineering allosteric sites into existing proteins has been carried out previously [344]. These efforts typically add an allosteric site into an existing functional protein or one adjoining an entire domain [345]. Full de novo allosteric proteins are still within the domain of classic design-and-check methods requiring expert attention at all steps [346]. Despite the success of some ML methods that rely on a single static protein structure, it has been our experience [16,34] that single mutations can often dramatically shift the communication pathways, yet the topological properties of the structure remain largely invariant. Accounting for fluctuations through rigidity networks or vibrational modes, or ensemble of conformations through sampling is likely required for robust results.

#### 3.4.3. Potential Energy and Force Field Calculations

To perform MD simulations, there must be a molecular forcefield, which must be parameterized. The most accurate method is using ab initio quantum calculations to determine the exact Hamiltonian of the system. Unfortunately, this direct approach becomes intractable for systems larger than several hundred atoms. In addition, the calculations are difficult in principle because environmental effects (from more atoms) are difficult to model implicitly, leading to a non-local parameterization that does not generalize to all systems. Alternatively, a functional form can be approximated based on modeling physical interactions at a local level (bonds, dihedrals, van der Waals, etc.) plus non-local electrostatics. These models can be parameterized based on fitting the functions to known experimental data. While this is more practical, it is limited by the experimental data available, which often does not generalize to systems and environments much beyond the realm described by the fitting data. Limitations from systematic experimental data availability will be present in all ML methods too, but ML may help the procedures to obtain better models for the data that are available.

In recent reviews of the subject [347,348], requirements for ML-based forcefield models are laid out. The main three requirements for the outputs of the model are: (1) conservative forces, which are (2) rotationally and translationally invariant, and (3) the interactions between particles respect symmetries regarding indistinguishable, identical atoms. Usually the nuclear charge allows the model to be aware of atom types. Most methods choose to predict potential energy functions instead of directly predicting forces. This is because potential energy functions are smoother and it is straightforward to differentiate potential energy functions to compute the forces from the model. Neural networks with rigid input structures are sensitive to not only these requirements, but also systems with variable numbers of atoms. Common approaches to this use permutations to match the input order to a universal form and transform the input system to faithfully provide the required invariance properties using transformations such as coulomb matrices or symmetry functions. Alternatively, many models use distance pairs as inputs to a model because they are intrinsically rotationally and translationally invariant. However, even in this case, additional transformations and features are needed to ensure that they are invariant upon atom permutation. The most problematic aspect of this is that this approach does not scale, and thus is not transferable, because the number of distance pairs makes the calculations intractable for large systems.

Ideally a molecular representation models an atom with its local environment, which influences the forces acting on the atom. To achieve this, as well as the other requirements for representations, Atom Centered Symmetry Functions (ACSFs) [349] were introduced in 2011. These functions had two parts: a cutoff function that identifies atoms in the proximity of the atom of interest, weighting contributions to the overall symmetry function, and a symmetry function which describes the local environment. Each ACSF is a sum over all distance pairs within the radius cutoff; hence, it is rotationally and translationally invariant and invariant upon exchange of atoms. ACSFs proposed in the original paper described the radial, spatial and frequency distribution of atoms in the local environment of the atom of interest but did not take into account the atom species. In later work, a set of weighted ACSFs was developed [350] that weighted each symmetry function contribution to the ACSF by the chemical species.

Several methods of transforming molecular representations for predicting forces with ML have been reviewed [351]. One of these is the Smooth Overlap of Atomic Positions (SOAP), which is a method for representing local atomic geometries as a continuous density for use in ML potentials. SOAP appears to be a superposition pairwise distance of atoms within each region, expanded on radial basis functions and spherical harmonics, then kernelized to describe similarity between environments. Another alternative to finding an invariant representation for the molecular system is to find the optimal aligned frame of reference to compute forces in. Unfortunately, this is not efficient for simulation applications, as forces will continually have to be transformed to the correct frame of reference [351].

The Born–Oppenheimer approximation allows the nuclear and electronic wave functions to be treated independently, where the nuclear degrees of freedom are treated classically. Because classical MD simulation does not simulate electron dynamics, a forcefield need not account for this detail. One approach developed in 2010 [352] is the Gaussian Approximation Potential (GAP). GAP first represents the local atomic environment in a set of Gaussian radial basis functions, and learns regression that predicts the energy for the atom using kernelized features. The Gaussian basis has the benefits of being intrinsically rotationally and translationally invariant, and is easily differentiated in terms of the atomic positions to provide forces [352]. In addition, GAP has been implemented in the QUIP software [353] for MD, which can be plugged into popular MD software such as LAMMPS. GAP appears to be fast and accurate for predicting configurational energy; however, it must be trained using ab initio calculations, which limits its generality.

Early work using ML to parameterize forcefields to simulate the dynamics of molecules involved using neural nets to predict parameterization for potential energy surfaces. Refs. [354,355,356,357] Neural Networks as potential energy surface approximators were explored further in 2004 [358]. Their models use a neural network to represent a system and output its total energy. These networks represented atomic configurations with symmetry functions, which drastically reduced the input size and allowed the networks to be trained using a reasonable number of data points. Behler and Parinello generalized this in 2007 [359] in their high-dimensional neural network (HDNN) model. The advantage of this model, which allowed it to generalize to higher dimensions, is that it broke the total energy calculation into the sum of energy contributions from each atom. The energy per atom was then computed from a set of subnetworks, one subnetwork representing each type of atom in the system. A detailed review and subsequent developments involving HDNNs can be found at [360].

More recent ML models have used the many advanced architectures and methods developed in the past few decades. PhysNet [361] is an example of a DNN architecture proposed for computing energy, force and dipole moments. CNNs have shown great promise in helping us understand the complex interactions and dynamics of proteins; however, the discrete, grid-like structure of convolution layers made them an unattractive solution for machine-learning-based forcefields. To accommodate the continuous nature of atomic coordinates within the CNN architecture, Schütt et al developed a continuous-filter convolution layer [362] (cf-Convolution) to transform atomic representations into feature representations more salient for energy prediction. The continuous-filter convolution takes in the feature value at layer *l*, $x_{i}^{l}$ and the 3D coordinates of each feature’s associated atom, $r_{i}$ and transforms the features a new representation in layer l+1.
(1)$$x_{i}^{l+1}=\Sigma_{j}x_{j}^{l}\cdot W^{l}\left|\right|r_{i}-r_{j}\left|\right|$$

The matrix *W* transforms the pairwise distances to the feature space dimensions. The model was implemented in a new network structure called SchNet [363] which has been implemented in a PyTorch toolkit called SchNetPack [364]. Forces on each atom can be extracted by following Newton’s laws and differentiating the predicted energy with respect to the atomic coordinates. The full SchNet model is similar to a fully connected GNN in that each atom is represented as a node connected by pairwise edges (displacements) and that the node representation is embedded into a *d*-dimensional feature space which is updated with each sequential cf-convolution layer added to the model. The difference is that convolutions, rather than message passing, are performed to update the feature representation. Recent trends have seen use of GNNs for molecular fingerprinting extensively explored for predicting molecular properties such as energy and force. Some examples of method that use GNNs for predicting energies and forces include DimeNet [365], GNNFF [366], and NewtonNet [367]. These methods have shown increased force prediction accuracy over other deep learning approaches such as PhysNet and SchNet.

There are several implementations of deep ML potentials that can be used in MD simulation software instead of classical potentials. TorchMD is an MD engine (downloadable via github) writen in pytorch that uses combined classical and deep learning potentials [368]. For TorchMD’s Deep Learning potential, SchNets were used. DeePMD-kit [369] is a method for parameterizing deep ML potentials, developed and distributed by deepmind. This can be installed via conda, and the resulting trained potential can be used in the LAMMPS and GROMACS simulations software.

### 3.5. Drug Design

One of the strongest drivers of ML method development for computational biology and protein design is drug discovery. The drug discovery process takes an average of 12 years from start to commercialization, with an average cost of USD 1.8 billion [370]; consequently, this necessitates a need to streamline the drug development process. To this end, ML approaches have been widely applied in the field of biomedicine. Typically, ML methods use pattern recognition algorithms to discern mathematical relationships between empirical observations, such as protein secondary structure prediction, drug repositioning, and drug design [371]. In recent years, deep learning has garnered considerable interest from computational chemists and medicinal chemists. Until now, various reviews related to the applications of ML or deep learning in drug design and discovery have been published [372,373,374,375,376,377,378,379,380]. Here, we focus on two vital computational components of the drug design/discovery process, molecular docking and binding affinity prediction, as highlighted in Figure 7.

#### 3.5.1. Molecular Docking

Molecular docking is a vital step in the drug design process, and acts as a filter to identify candidate molecules for the purpose of reducing costs by streamlining experimental work. Virtual screening often refers to a high-throughput candidate selection procedure [381]. The pairwise objective with molecular docking can include: protein–peptide, protein–DNA, protein–RNA, and protein–ligand. Depending on the method, 3D structures, physiochemical properties or a combination of the two are utilized to predict binding characteristics [382]. Molecular docking is a combination of two processes. The first is sampling, which involves generating a set of conformations from a semi-rigid 3D ligand. The method is evaluated based on its capacity to explore the conformational space of the ligand (target), gathering all theoretically possible conformations [383]. The second step is scoring, which approximates the binding affinity of each protein–ligand complex formed (called a pose). Other methods use scaffolds, or molecular fields that represent molecules to make the coarse-grained exploration of conformational space more computationally efficient [384]. Molecular docking can not only predict the binding conformation of a ligand in a target binding pocket, but can also estimate the binding affinity of ligand–target complex [371]; the latter will be expanded upon in the following section.

Classical docking scoring functions assume an additive functional form in order to model a linear relationship between the binding affinity and features characterizing protein–ligand complexes [385]. Unfortunately, a linear relationship is not guaranteed to exist; furthermore, if a non-linear method is used to fit this relationship, the developed scoring function could attain better performance. Traditional ML methods such as random forest [386], SVM [260,387] and neural networks [388,389] have been utilized for molecular docking scoring in lieu of regression-based approaches. Recent algorithmic advances in deep learning have driven promising molecular docking approaches. These deep learning docking methods come in many forms, such as: CNN [388,390,391,392], GNN [393,394,395,396], and generative models (GANs/VAEs) [321,397,398,399].

We identify three considerations that should be taken into account which ground many of the remaining challenges in molecular docking. First, not all the complexes in the PDB are functional, and this consideration should be reflected when determining protein sets for training ML models [400]. Furthermore, when developing ML models for molecular docking, it is important to train and validate the models over established data sets instead of using synthetic or augmented data sets. This guarantees representativeness, exhaustiveness, variety for the training set, and allows for inter-method comparisons of objective criteria [383]. Second, a general trend is that on average, the size of the interface measured as solvent accessible area buried upon complex formation is larger in biological interfaces compared to crystallographic structures [401]. Lastly, and possibly most important, the three-dimensional structure used for molecular docking will be out of its original environment, often resulting in a change in conformation; thus, the docking result cannot truly reflect the state of the experimental docking [402].

#### 3.5.2. Binding Affinity

Binding affinities are important for evaluating novel drug molecules and their targets. Within computational biology, this falls under applications of virtual screening and docking. The dissociation constant, $K_{d}$, is typically used to find binding affinity, as the two quantities are inversely proportional. Other useful metrics, particularly for enzymes and inhibitors, are the inhibition constant $K_{i}$, the half maximal inhibitory concentration (IC50) and the minimum inhibitory concentration (MIC). For high-throughput applications such as virtual screening, the measurement of thousands of $K_{d}$ is inefficient, so there is much motivation for computationally fast and accurate predictions of binding affinities. In molecular docking, methods refer to binding affinity or just affinity as the scoring function for a particular ligand conformer. While some of these models can be based on physical interactions, and the physical binding affinity is certainly dependent on protein–ligand or protein–protein interactions, the predicted affinity is often not physical, and these scores are used primarily as a relative measure that should linearly correlate to true binding affinities in the best cases.

In principle, ML can be trained on experimental data to directly predict binding affinities. Therefore, datasets are important for this approach. A few examples of databases which contain binding affinity information are BioLiP [403], Binding MOAD [404], BindingDB [405], and PDBbind [406,407]. An early example of how ML can accelerate binding affinity calculations was the RF-Score [408], which used RF models. RF-Score trained on distance pair occurrences between types of protein and ligand atoms, and achieved state-of-the-art performances when it was released in 2010. Random forest ML improved classic binding affinity calculations by training models on features produced by models such as CYSCORE or AutoDock Vina [409,410]. Head-to-head comparisons between classic and ML prediction algorithms [411] have been performed, showing that ML models can outperform classic models in most useful tasks, such as predicting, ranking and docking with binding affinities. Finally, among traditional non-deep ML methods, RF is often the best predictor [412].

With deep learning and large datasets becoming increasingly accessible, various neural network architectures have been exploited, particularly the CNN architecture. Early work on applying CNNs to predict binding affinity was carried out in 2009 [413], where it was found that CNN provided highly accurate affinities for cations binding to common amino acids. In 2017, the subject of using CNNs as scoring functions for virtual screening and binding affinity prediction was picked up [414], and this became the dominant model. To prepare protein ligand structures into the correct input shape for a CNN, the interaction surface is often turned into a 3D grid, with each pixel representing information about the atoms contained within it. This type of CNN that learns directly from the 3D structure of the protein ligand complex is called a 3D CNN. The 3D input structure allows the input to include information about the local environment of the ligand. KDEEP [415], Pafnucy [416], and DeepAtom [417] are examples of a CNN that utilizes these 3D convolutions. While 3D CNNs are powerful predictors, it has been noted that moving from 2D to 3D drastically increases parameter space, allowing for more possibilities of over-fitting and slowing down the prediction time. This was addressed by DeepBindRG [418], which deflated the input size to a 2D array, rather than the 4D array needed for 3D CNNs, and used ResNet for binding affinity prediction. Another popular approach is OnionNet [419,420], which featurizes protein–ligand complexes in a similar way as RF-Score, using atom-pair-specific contacts but at varying spatial scales to create a 3D input array appropriate for 2D convolution learning. There is also work on applying CNNs directly to complexes in sequence space in DeepDTA [421], where the protein is represented by its sequence and the ligand by its SMILES code. The two representations are processed by independent CNN blocks and fed into a single CNN block that predicts the binding affinity. It is worth noting that DeepDTA has been highly influential in binding affinity prediction, with many models directly building off it.

Other deep-learning approaches have been used to predict binding affinities. For example, Zhao et al. [422] replaced the CNNs in the feature extraction blocks of DeepDTA with a set of GANs. With this modification, the GAN learns to generate features in an unsupervised manner. This semi-supervised framework had similar performance to its predecessor, but could use more data, which greatly reduces the need for labeled training data. Again, by building off DeepDTA’s sequence approach, attention layers have been shown to increase the effectiveness of the model [423] while making it interpretable and allowing it to predict where binding sites might be in sequence space, with variable accuracy. Finally, GNNs have become popular for feature extraction and prediction tasks. GNNs have previously been shown to create “molecular fingerprints” that act as useful representations of molecules. Two methods which have used GNNs for are GraphDTA and GraphBar [424,425], both of which use GNNs to directly compute binding affinity. GraphDTA is similar to DeepDTA, but the CNN that featurizes the ligand is replaced with a GNN. GraphBar takes a hierarchical approach by constructing multiple graph representations of the complex using increasing distance cutoffs for interactions.

Another random forest model called iSEE predicts changes to binding affinity, ΔΔG, due to mutations [245]. Again, databases are needed to train ML models. One such database is SKEMPI, which lists ΔΔGs in response to many mutations that can be used for ML training [426]. The iSEE paper compares to several other ΔΔG predictors including FoldX [427], CC-PBSA [428], BeAtMuSiC [429], BindProfX [430], which uses evolutionary information from homologs, and mCSM [431,432]. To account for the effects of mutation, either MD simulations for conformational sampling and evolutionary sequence information in terms of models such as Position Specific Scoring Matrices (PSSM) must be exploited to predict changes to binding energies.

Although molecular docking and prediction of binding affinity is a mature area of computational biology that has been impactful in drug design for more than two decades, there remain open challenges. Better methods are needed to robustly identify binding sites on protein structure. This proved to be a difficult task, as binding models must consider environmental and preferential binding effects due to thermodynamic linkage [433,434,435,436]. Binding affinity includes thermodynamic and mechanical stability concerns involving enthalpy–entropy compensation, which leads to observable cooperativeness. To accurately model binding affinity, dynamic aspects of the protein and its binding partner must be taken into account beyond local flexibility characteristics. The result of these complexities is that scoring functions typically prove inadequate, because entropic effects are difficult to quantify, and capturing these effects requires considerable exploration of conformation ensembles. Although the experimental side of high-throughput screening is time consuming and costly, it is also the case that simulation of conformational ensembles is also costly and time consuming. We believe that methods that account for conformational ensembles and entropy will dominate the field in the long run, with the remaining methods becoming obsolete, even if they perform best in the short term for limited data sets. Including dynamics and environmental effects is essential to make non-incremental progress in this area, which amounts to combining the four pillars of computational biology.

## 4. Conclusions

Our review cited over 300 papers on ML methods used in applications of computational biology, and over 100 papers on protein function and computational methods that have historically been successful in protein function analysis. We tied together the ML methods used to elucidate protein function, along with practical applications involving proteins with drug discovery. The context of the underlying biophysics, biochemistry and molecular biology puts into perspective the domain knowledge required to successfully integrate ML into computational biology applications. The convergence of computational biology with ML has already shown many fruitful directions. Due to the complexity of the subject, we believe that the recent gains in model predictions for protein function or functional attributes will continue to increase in the foreseeable future. We have emphasized the conformational ensemble perspective, because there is a natural continuum that bridges proteins with various degrees of conserved structural motifs and disorder. We believe that modeling conformational ensembles and protein dynamics under different environmental conditions is necessary to make the most progress in protein science through computational biology. It is the modeling of these general concepts that also pushes ML to its current state-of-the-art limits. The convergence of computational biology with ML is clearly beneficial for both areas of study, as challenges in protein function analysis are addressed.

### Future Opportunities

Despite recent work over the last two decades in terms of accurately describing conformational ensembles of proteins, more work is needed to represent these ensembles in a computationally tractable and mathematically complete way. ML is a tool that can help achieve controllable approximation. We stressed that protein function is linked to specificity, because proteins function in the crowded cellular environment. Understanding how proteins function under environmental changes requires finding representations for environmental effects. It appears ML can improve protein engineering through in silico mutation studies. Changes in the environment and the primary structure of a protein together have significant effects on stability and function, both of which are related to the conformational ensemble.

Another necessary step is to find ways to simulate protein dynamics over long time scales. Due to the intrinsic limitation of the MD methodology of integrating differential equations on the femtosecond time scale, coupled with the problem that biologically important timescales routinely exceed one second, and the desire to simulate ever larger systems, there will always be a major problem with sampling on classical computers in the foreseeable future. Although this sampling problem presents a bleak picture, there are already many ways to address this problem by using coarse-grained models, metadynamics, and other bias techniques to explore functionally relevant dynamics, with ML greatly improving the effectiveness of these methods. There is also the challenge of identifying functional dynamics, for which ML can be used for discriminant analysis and dimensionality reduction of MD simulation data. On these fronts, ML provides a means to solve—or at least mitigate—sampling problems and identify the functionally relevant part of the generated conformational ensemble. To obtain accurate thermodynamic properties, such as binding affinity, it is imperative that the appropriate conformational ensemble is sufficiently sampled. Although more sampling allows flexibility in docking to be accounted for, the docking application remains a challenging problem.

The sampling challenge also motivates the ongoing development of better forcefields for MD simulations and thermodynamic models that take into account non-additivity in conformational entropy. We also reviewed progress on forcefield development using ML methods that have already made progress on these challenges and promise many more advances in the near future. In this time of convergence between ML and computational biology, decades of nurtured ideas have begun to bear fruit. The protein folding problem, while remaining a major challenge in protein science, is on the cusp of being solved, as are many other grand challenges in protein science.

## Figures and Tables

**Figure 1 biomolecules-12-01246-f001:**
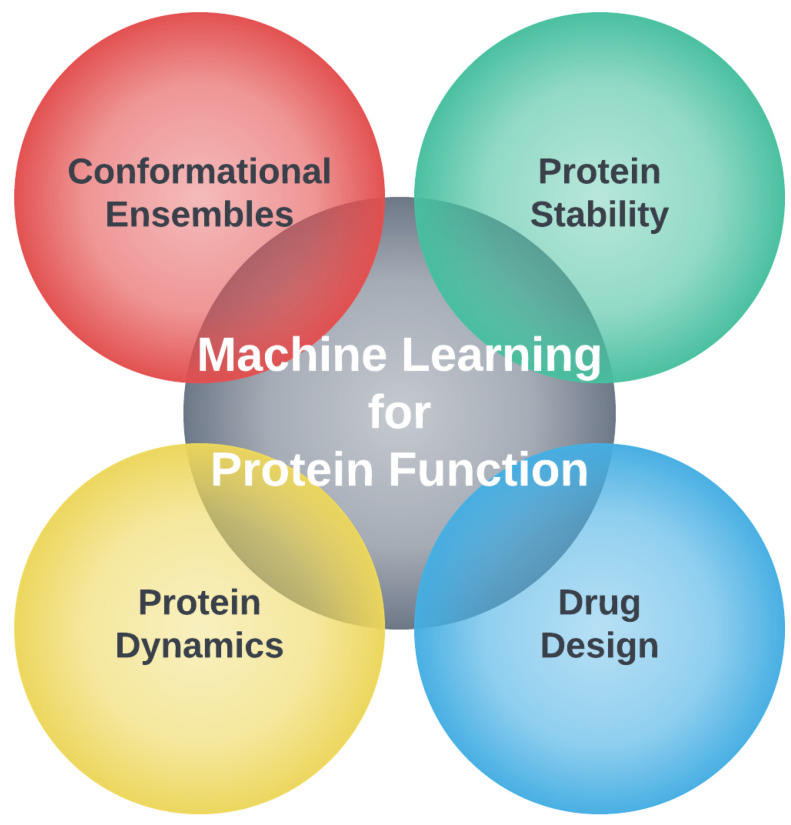
Machine learning meets computational biology. This review is about how machine learning (gray center circle) intersects with multiple aspects of computational biology (colored circles).

**Figure 2 biomolecules-12-01246-f002:**
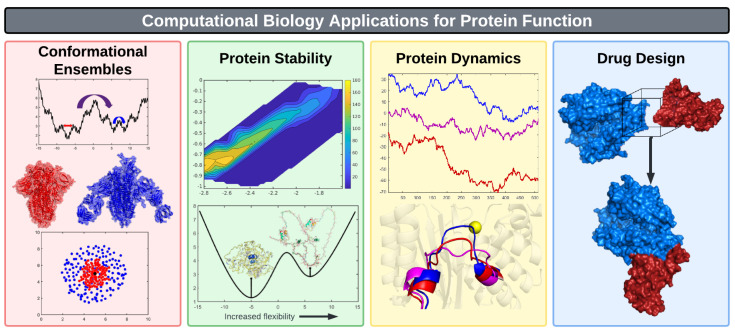
Four pillars of computational biology: The role of protein structure is critical in all panels. Conformational ensembles, red panel: ML helps with: (*i*) enhanced sampling; (*ii*) identifying collective variables; (*iii*) automated potential biasing; and (*iv*) Markovian state space partitioning. Protein stability, green panel: ML helps with (*i*) modeling the role of environment; (*ii*) protein engineering through mutagenesis; (*iii*) characterization of protein–protein interactions; and (*iv*) the role of rigidity for protein function. Protein dynamics, yellow panel: ML helps with (*i*) protein flexibility/conformational dynamics; (*ii*) dynamic allostery; and (*iii*) potential energy/force fields. Drug discovery, blue panel: ML helps with (*i*) molecular docking; and (*ii*) binding affinity prediction.

**Figure 3 biomolecules-12-01246-f003:**
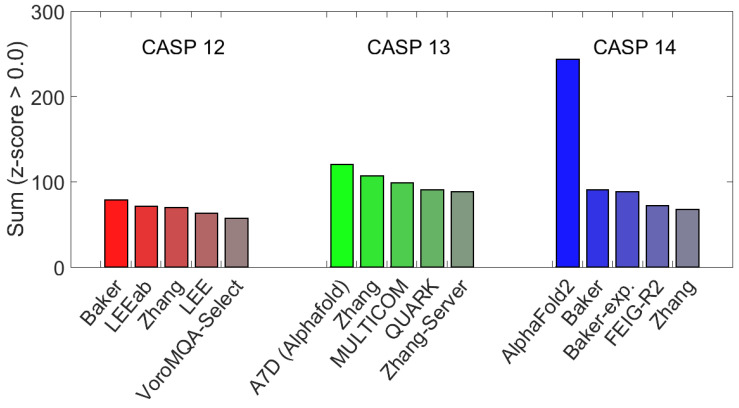
CASP 12-14 top 5 competitors per year (x-axis). The performance for each competition was based primarily on the summation of positive Zscores (y-axis) with respect to GDT_TS for each of the proposed structure prediction models. The accuracy metric, GDT_TS, is a multiscale indicator for the proximity of Cα atoms in a model to those in the corresponding experimental structure.

**Figure 4 biomolecules-12-01246-f004:**
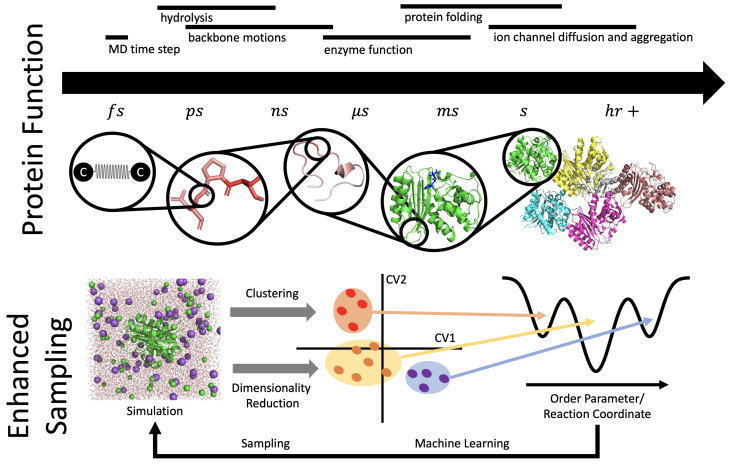
Time scales of protein function. Protein function occurs on time scales that span many orders of magnitude. MD simulation time steps are limited to the femtosecond range, necessitating enhanced sampling methods for the analysis of long time scale processes. Enhanced sampling typically starts with an unsupervised simulation of proteins to initially explore conformation space. Clustering and dimensionality reduction inform methods such as metadynamics on how to bias further simulations for the exploration of unsampled or poorly sampled regions in the free-energy landscape. Machine learning has been deployed to achieve better and faster sampling of this landscape.

**Figure 5 biomolecules-12-01246-f005:**
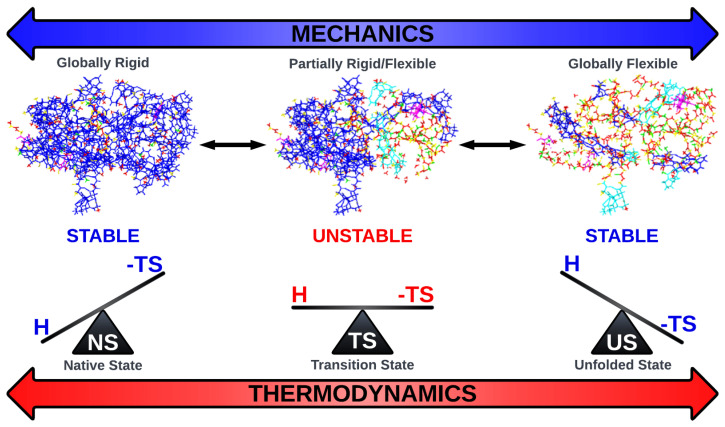
Protein stability paradigm for globular proteins. Mechanical and thermodynamic stability are intimately related. The native state of globular proteins is driven by favorable enthalpic interactions through the hydrogen bond network and packing interactions. The unfolded state is driven by conformational entropy, associated with an increase in conformational flexibility. The transition state represents a mixture of opposing thermodynamic and mechanical elements, which determines protein folding pathways.

**Figure 6 biomolecules-12-01246-f006:**
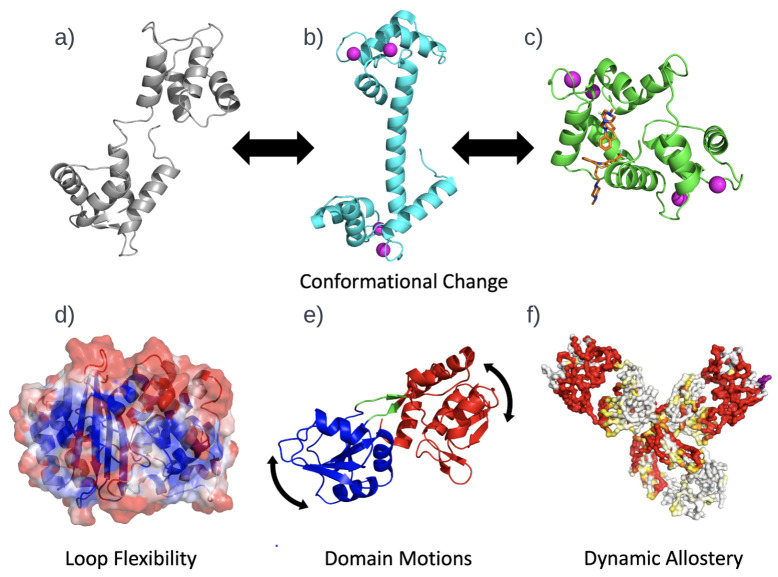
Examples of protein functional dynamics. In (**a**–**c**), three critical conformational states [297,298,299] of calmodulin are shown during the process of ligand binding. The unbound structure (**a**) binds to calcium ions (**b**), then the resulting structure is able to bind with a substrate (**c**) [300]. On the bottom row, (**d**,**e**) show the native state motions of proteins including surface loop flexibility [301,302] and concerted domain fluctuations [303]. (**f**) A visual of dynamic allostery pathways resulting from changes to the vibrational modes of a protein upon binding a ligand [304].

**Figure 7 biomolecules-12-01246-f007:**
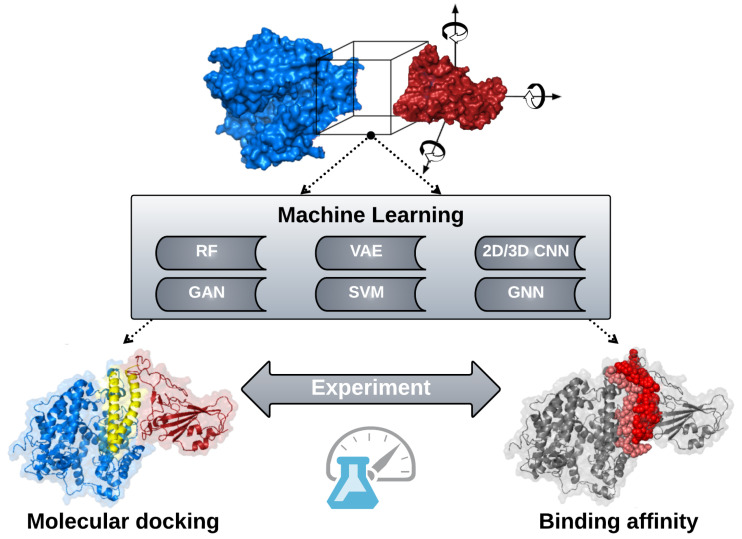
Computational drug discovery relies on the interplay between molecular docking and binding affinity prediction, both of which have been enhanced by ML. Docking methods can use ML to account for subtleties in molecular interaction such as flexibility or predict inter-molecular contacts, while ML-powered binding affinity functions score poses. In both cases, experimental data are used to train models, and methods will continue to improve as more data are collected.

**Table 1 biomolecules-12-01246-t001:** Survey of commonly used machine learning models. Abbreviations defined here are used throughout this review.

Method	Task	Paradigm	Abbreviation
k-Means	clustering	Unsupervised	-
Agglomerative Clustering	clustering	Unsupervised	-
Spectral Clustering	clustering	Unsupervised	-
Self-Organizing Maps	clustering	Unsupervised	SOM
Principal Component Analysis	DR	Unsupervised	PCA
Time-Lagged Independent Component Analysis	DR	Unsupervised	tICA
t-Distributed Stochastic Neighbor Embedding	DR	Unsupervised	t-SNE
Supervised Projection Learning for Orthogonal Completeness	clustering/classification/DR	Unsupervised	SPLOC
Naive Bayes	classification/regression	Supervised	NB
Support Vector Machines	classification	Supervised	SVM
Decision Trees	classification/regression	Supervised	DT
Random Forest	classification/regression	Supervised	RF
Gaussian Mixture Model	classification/regression	Supervised	GMM
Artificial Neural Network	classification/regression/NLP	Supervised	ANN
Shallow Neural Network	classification/regression/NLP	Supervised	SNN
Deep Neural Network	classification/regression/NLP	Supervised	DNN
Convolutional Neural Network	classification/regression/NLP	Supervised	CNN
3D Convolutional Neural Network	classification/regression/NLP	Supervised	3D-CNN
Recurrent Neural Network	classification/regression/NLP	Supervised	RNN
Autoencoder	DR/generative modeling	Supervised	-
Variational Autoencoder	DR/generative modeling	Supervised	VAE
Generative Adversarial Network	classification/generative modeling	Supervised	GAN
Message Passing Neural Network	classification/regression/DR	Supervised	MPNN
Graph Neural Network	classification/regression/DR	Supervised	GNN
Graph Convolutional Neural Network	classification/regression/DR	Supervised	GCNN

**Table 2 biomolecules-12-01246-t002:** Top ML methods for PSP in the regular targets category from CASP12 to CASP14.

CASP	Method/Group	Year	Model ${}^{1}$	Stand Alone ${}^{2}$	Webserver ${}^{3}$
12	Baker [75]	2016	Rosetta	Yes	Yes
	LEEab [76]	2016	GOAL	No	No
	Zhang [77]	2016	I-TASSER	Yes	Yes
	LEE [76]	2016	GOAL	No	No
	VoroMQA-select [78]	2016	VoroMQA	Yes	Yes
13	A7D (AlphaFold) [79,80]	2018	FreeM + DNN	Yes	No
	Zhang [81]	2018	TripletRes	Yes	Yes
	MULTICOM [82]	2018	CDNN + ab initio	Yes	Yes
	QUARK [83]	2018	C-QUARK	Yes	Yes
	Zhang-Server [83]	2018	C-I-TASSER	Yes	Yes
14	AlphaFold2 [84]	2020	TR/ATT + ANN	Yes	Yes
	Baker [85]	2020	ResNet + trRosetta	Yes	Yes
	Baker-exp. [86]	2020	ResNet + trRosetta	Yes	Yes
	FEIG-R2 [87]	2020	PREFMD2	Yes	Yes
	Zhang [88]	2020	UnknownI ${}^{4}$	No ${}^{4}$	No ${}^{4}$

^1^ This column lists the name of the model used by the CASP competition group. Due to the extensive architectures used, some models do not have ML shorthand nomenclature. In-depth descriptions of the models can be found in the corresponding reference. ^2^ This column indicates whether a standalone program is available (“Yes” or “No”). The hyperlink for “Yes” redirects to the corresponding repository. ^3^ This column indicates whether a webserver is available to users (“Yes” or “No”). The hyperlink for “Yes” redirects to the corresponding webserver URL. ^4^ The exact model was ambiguous, as this lab submitted multiple high-performing models.

## Data Availability

Not applicable.

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
