# Peer review of "Protein Function Analysis through Machine Learning"

_biomolecules, 2022, doi:10.3390/biom12091246_

Round 1
Reviewer 1 Report
Avery et al. reported an overview of computational methods for different applications related to prediction and analysis of proteins structures, dynamics and function. Such an overview would be very beneficial to the field. However, while the content would be of high interest, major concerns are raised in respect to the content, style and correctness of information of the manuscript. In the current version, the work is not recommended for publication. The phrasing of the manuscript is not accurate and far from using appropriate terminology typical of the field. This brings confusion and unclarify in key concepts, misleading information and difficult comprehension overall of the text. Just a few examples are reported here: Page 3 line 98: "The strength of a protein's interaction with itself". Unclear what the authors mean here... self-assemblies? Page 3 line 117: "with corresponding amino acids". The authors should introduce the concept of sequence alignment and describe accordingly. The "corresponding amino acids" is prone to any interpretation and confuse the information they try to provide. Page 3 line 124: "subtle differences". What are the 'subtle differences' in proteins? Do the authors mean amino acids mutations? Page 3 line 125: "for quantity protein function" is really difficult to comprehend here. What exactly would be quantified? Page 4 line 151-152: "how proteins fold dynamically, what would they mean here? Page 4 line 192-193: "predicting protein-protein interactions involves a docking problem that is particularly difficult because of the size of the binding partners". This is way completely inaccurate. The problem of protein-protein docking ranges from conformational changes upon complex formation which is difficult to predict, properly sampling conformational space, properly scoring and ranking docked poses... Those are just some of many incorrectness reported all over the manuscript.Author Response
We appreciate the general comments of reviewer 1, which was on point. We would like to explain why the initial submission was under par. We asked the academic editor of the special issue if we could get an extension, but this was not possible. We did the best we could to make the deadline, and what was submitted was less than a normal first draft. We did not have a chance to blend together the different sections that each of the authors were responsible for. We had no time to finish the planned figures. We are sorry for this. The background search/reading spent on this review was considerable.
This revision is a polished version. Blue highlights show major changes. Other minor changes were not always highlighted. We took into account the specific suggestions of reviewer 1, but obviously it can be seen by the blue highlights that a MAJOR revision has been done. We hope this version will be ready for publication, although this was our intended version we wanted to submit. Any constructive criticism will be appreciated.
Reviewer 2 Report
This is a very comprehensive work on ML in the field of computational structural biology. I believe that this work should be accepted, pending a few moderate to significant revisions.
1) Proofreading is necessary. The English is fairly comprehensible, but it is sometimes poor. Please add the appropriate citations that have been left as "?" in the text. I am a bit taken aback that the authors would have submitted an article with incomplete citations.
2) In section 2.2 where the application of ML to computational biology is introduced, there is no discussion of supervised vs. unsupervised ML. This is the principal distinguishing feature of ML methods. Although this distinction is made later on in the text, it must be introduced at this stage.
3) The review is very comprehensive, but all too often it reads like a list of methods with almost no discussion of the relative performance of the various ML-based methods. The reader is left bewildered as to when to apply one method over another. There is no organized discussion of what the ML methods get right, as opposed to purely first-principles-based or physics-based methods.
Author Response
We appreciate the suggestions of reviewer 2, which has made our review much better. We would like to explain why the initial submission was under par. We asked the academic editor of the special issue if we could get an extension, but this was not possible. We did the best we could to make the deadline, and what was submitted was less than a normal first draft. We did not have a chance to blend together the different sections that each of the authors were responsible for. We had no time to finish the planned figures. We are sorry for this. The background search/reading spent on this review was considerable.
Blue highlights show major changes. Other minor changes were not always highlighted.
We appreciate you overlooking the mess we had in our first submission due to our being pressured in meeting the initial submission deadline. You provided us with great feedback. We added in a considerable amount of material giving an overview of ML methods in the ML subsection as suggested. This was a really good idea. Thank you.
Originally, we were not going to provide some words of recommendations, but we now have provided recommendation in almost all sections, if not all. These recommendations usually are cautionary in nature.